

# Optimizing rock glaciers activity classification in South Tyrol (North-East Italy): integrating multisource data with statistical modelling

Chiara Crippa [1], Stefan Steger[2], Giovanni Cuozzo[1], Francesca Bearzot[3], Volkmar Mair[4],
Claudia Notarnicola[1]

[1] Institute for Earth Observation, European Academy of Bozen / Bolzano, Eurac Research, Italy
[2] GeoSphere Austria, Vienna, Austria
[3] Faculty of Earth, Energy, and Environment, University of Calgary, Canada
[4] Office for Geology and Building Materials Testing, Autonomous Province of Bolzano-South Tyrol, Cardano, Italy

*Correspondence to*: Chiara Crippa (chiara.crippa@eurac.edu)

**Abstract.**

As a consequence of climate warming, high-altitude periglacial and glacial environments exhibit the clearest signs of cryosphere degradation, and the Alps serve as a natural laboratory for studying the primary effects on permafrost-related features. Our research in South Tyrol, North-East Italy, aimed to develop an updated classification system for rock glaciers activity, based on remote sensing data and statistical models, with the aim of categorizing them as active, transitional, or relict according to the recent RGIK guidelines. Since the current regional inventory includes activity attributes based only on morphological observations and differential SAR interferometry (DInSAR) coherence, it lacks a comprehensive definition integrating climatic drivers, displacement rates, and morphometric parameters. To address this, we utilized the Alaska Satellite Facility's InSAR cloud computing, employing small baseline subset (SBAS) approach and MintPy algorithms to extract velocity data for each rock glacier in South Tyrol. Additionally, we analyzed geomorphological and climatic maps derived from in-situ and remote sensing data to obtain descriptive parameters influencing rock glaciers development and activity. From a wide range of potential variables, we selected eight key predictors, representing physical (e.g. temperature), morphological (e.g. roughness), and dynamic (e.g. velocity and coherence indicators) attributes. These predictors were successively integrated in a multiclass generalized additive mixing model (GAM) classifier to categorize the landforms. Applying this model to the entire dataset (achieving an AUC over 0.9) allowed us to address gaps in previous classification methods and provided activity attributes for previously unclassified rock glaciers, along with associated uncertainty values. Our approach improved classification accuracy, leaving only 3.5% of features unclassified compared to 13% in morphological classification and 18.5% in DInSAR-based methods. The results revealed a predominance of relict features (~75%) and a smaller number of active ones (~10%). The distribution of active, transitional, and relict classes suggests that the transition from active to relict states is not a direct process. Instead, an intermediate transitional phase is commonly observed. This comprehensive approach refines the categorization of mapped features and improves our understanding of the factors influencing rock glaciers activity in alpine environment.



## 1 Introduction

Rock glaciers are widespread periglacial landforms in mountain regions and are regarded as key geomorphological evidence of permafrost presence in alpine environments (Haeberli, 2000). They consist of a continuous, thick seasonally frozen debris layer (known as active layer), covering ice-supersaturated debris or pure ice. They are characterized by gravity-driven creep as a consequence of ice/debris mixtures deformations under permafrost conditions (Haeberli et al., 2006), which promote a distinctive surface topography (i.e., ridges and furrows complexes, convex transverse or longitudinal surface undulations).

The large-scale spatial distribution of rock glaciers is influenced by the complex interaction of topographic factors and climate, specifically by mean annual air temperature and precipitation. However, on a local scale, their distribution is dictated by local factors such as slope and aspect, structure and lithology of bedrock, debris input, heat budget of the ground, shading, and duration and thickness of snow cover (Cicoira et al., 2019; Kenner and Magnusson, 2017; Bodin et al., 2009). Rock glaciers distribution and evolution, and current permafrost degradation may affect the slope stability, runoff patterns,

vegetation coverage, and water availability and quality, promoting landslides, geological disasters, debris flows, destabilization phenomena (Pruessner et al., 2021; Marcer et al., 2019; Gruber and Haeberli, 2007), and direct or indirect risk to human activities and/or facilities (e.g., infrastructures, buildings) (Hassan et al., 2021; Arenson and Jakob, 2017). Furthermore, some rock glaciers act as essential hydrological reserve in high mountain systems, prolonging long-term water (and ice) storage, and consequently their presence and abundance could affect the amount and properties of runoff from high

mountain watersheds (Bearzot et al., 2023; Wagner et al., 2021; Brighenti et al., 2019).

The genesis of rock glaciers has been debated for a very long time during which some studies claim the relationship between rock glaciers to periglacial conditions and permafrost presence (Knight et al., 2019; Haeberli et al., 2006), or linked them to paraglacial processes (Frauenfelder and Kääb, 2000) whereas others suggest the glacial origin of rock glaciers (Monnier et al., 2013; Krainer and Mostler, 2000; Whalley and Palmer, 1998) in which they originate from the evolution of debris-

covered glacier and where interstitial ice is glacial origin rather than meteoric (i.e., permafrost). Depending on their permafrost content and activity, rock glaciers have been categorized into three categories: (i) active rock glaciers, in which the internal deformation of frozen material and ice produces an effective surface displacement, (ii) inactive (dynamic or climatic) rock glaciers that still contain ice but have stopped moving and (iii) relict rock glaciers that no longer contain ice and consequently with no movement (RGIK, 2023). The active and inactive rock glaciers are commonly grouped together

into the class called intact rock glaciers. Although widely used, this classification brings two relevant limitations both from subjectivity point of view (activity attribution based on geomorphological approach is depended on the operator skills) as well as categorization since the activity of rock glaciers is considered constant over time at the scale of decades to centuries. In response to the ongoing increase in permafrost temperature, an acceleration trend has been observed worldwide, although with different phases based on the geographical regions and the characteristics of the individual landforms. For these

reasons, the existing rock glaciers classification was redefined as follow: (i) active rock glaciers (A) which moving downslope over most of its surface and present steep front and lateral margins contain freshly exposed material on top, (ii)



transitional rock glaciers (T) which show slow movement to no downslope movement over most of its surface and can either evolve towards a relict or an active state, depending on topographic and climatic context and (iii) relict rock glaciers (R) that show no evidence of recent movement, generally characterized by smoothed lateral and frontal margins and by the presence

of vegetation and soil cover (RGIK, 2023). Therefore, this updated classification does not consider the ground ice content but rather the efficiency of sediment conveyance, namely the surface movement at the time of observations.

In the regional territory of South Tyrol (Eastern Italian Alps), two rock glaciers activity classifications coexist over the same inventory, one is the South Tyrol Inventory produced by the Autonomous Province of Bolzano/Bozen (PAB) and the other one is made by Bertone et al. (2019). Although a descriptive attribute of activity from independent morphological

observations and a SAR coherence-based estimation is already included in the two datasets, a comprehensive definition of activity based on the integration of climatic drivers, displacement rates, and morphological parameters is lacking.

The primary innovation of this study lies in the analysis of multiple variables each one describing a key evidence or predisposing condition of rock glaciers activity, integrated through multivariate statistical analysis in a predictive multiclass generalized additive mixing model (GAM). We extracted the input variables by leveraging diverse sources, including

multispectral (Landsat, MODIS) and radar (Sentinel 1) satellites, interpolated ground measurements (weather stations) and digital terrain model (DTM) to derive morphometric factors.

To this aim we propose a workflow where i) we first exploit satellite remote sensing products and implement routines to extract velocity attributes and environmental descriptors at the regional scale; ii) we then calibrate and validate predictive multiclass GAM that maximizes their explanatory potential; iii) we apply the model to the entire dataset reclassifying each

landform in a specific activity class. Our approach effectively highlights which variables (such as climatic, morphological, and dynamic parameters) and interactions best control each rock glaciers' class of activity in the area investigated. Throughout this paper, the recent classification (A, T and R classes) was considered to define the activity of rock glaciers.

## 2 Study area

The study area covers the entire South Tyrol region (Northeast Italy, ~7400 km$^2$) and extends over altitudes between

approximately 200 m a.s.l. in the valley bottoms to 3900 m a.s.l. of Ortler Peak. The Periadriatic Line (P.L.; Fig.1a) separates the central eastern part, where sedimentary and metasedimentary rocks of the South Alpine domain outcrop, from the western regions characterized by the metamorphic lithologies of the Austroalpine and Pennidic domain, outcropping in the north easternmost sector (Stingl and Mair, 2005). The climate of South Tyrol is characterized by a rather continental character, with mean annual precipitation sum (period 1981-2010; Crespi et al., 2021) generally around 1000 mm. However,

the precipitation varies largely in South Tyrol from a regional point of view: the western sector, which includes Val Venosta (Fig.1b) and its side valleys such as Val Senales, Val di Trafoi, Val Martello, and Val d'Ultimo, has less precipitation (average annual precipitation ≤ 825.2 mm) than the central and eastern sector, which includes the vast highland in central and eastern South Tyrol (average annual precipitation > 825.2 mm) (Hao et al., 2019). Mean annual temperature extracted



over the same period 1981-2010, are around 12°C in the valley bottoms and decrease on the slopes till reaching the 0°C
isotherm around at 2400-2500 m a.s.l. (Crespi et al., 2021; Carturan et al., 2023).

Regarding the permafrost map (www.provincia.bz.it/edilizia/progettazione/alto-adige.asp), the region is characterized by
discontinuous mountain permafrost which develops from a minimum height of 2300-2400 to 2500 m. a.s.l. (Fig.1b),
according to sectors and site-specific climate conditions (Boeckli et al., 2012).

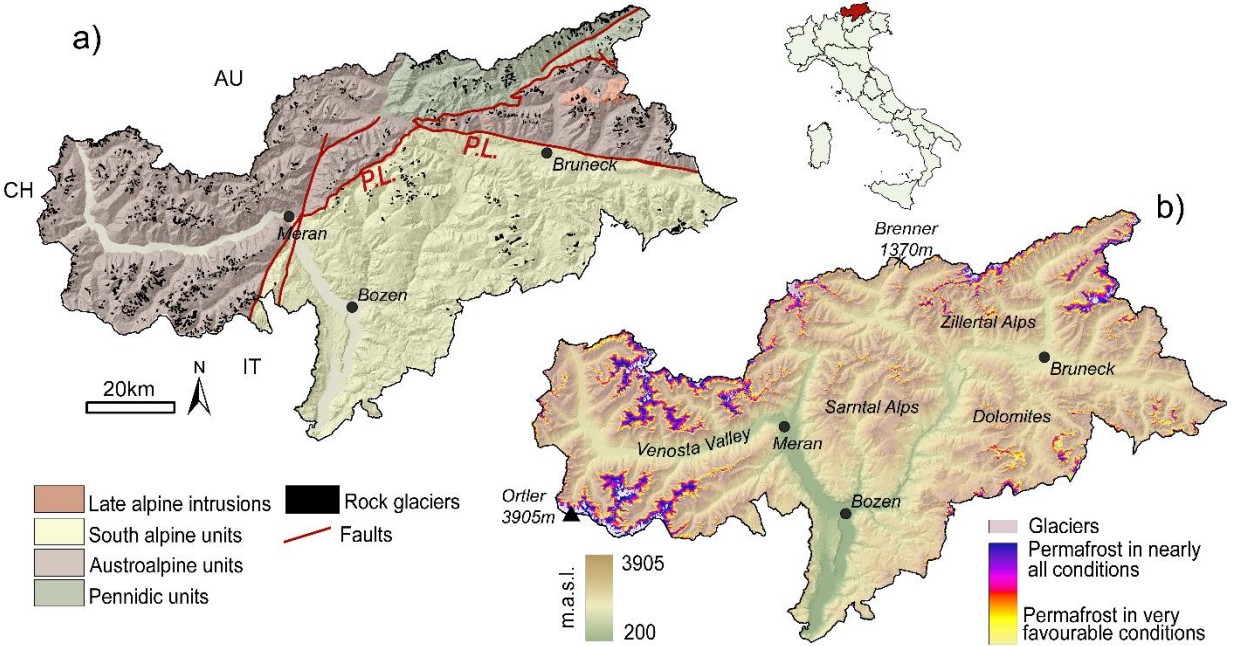

**Figure 1: South Tyrol region: a) lithological and structural map of the main geological units and faults in South Tyrol.
Rock glaciers of the autonomous province of Bozen dataset are highlighted in black; b) digital elevation model with
permafrost and glaciers distribution.**

## 3 Data collection and analysis

Multi source and multi platforms data from remote sensing products and ground-based measurements were collected and
jointly analysed. Using MODIS and Landsat satellite data allows the extraction of environmental parameters such as snow
cover duration and land surface temperature. MODIS, on board of Terra and Aqua satellites, with its multispectral
capabilities and daily repeat time, demonstrated efficacy in extracting the snow cover area both regionally and globally
(Notarnicola, 2020). Using Landsat 8 (nominal spatial resolution 100 m) and operational line imager and thermal infrared
Sensor (OLI and TIRS) satellite data, we extracted Land surface temperature (LST) which has been acknowledged as one of
the Essential Climate Variables (ECVs) by both the Global Climate Observing System (GCOS) and the Climate Change
Initiative (CCI) of the European Space Agency (ESA) (Galve et al., 2022; Parastatidis et al., 2017; Ermida et al., 2020).



Sentinel-1 SAR images were processed using HyP3 software (section 2.3) to retrieve dynamic attributes i.e., velocity and coherence values over the entire area of interest (AOI).

These variables, coupled with a range of data pertaining to rock glaciers morphometry and encompassing factors such as slope angle, elevation, lithology, and climate conditions, lead to a redefinition of activity classifications for all the mapped landforms within the pre-existing dataset. The overall workflow is sketched in Fig. 2.

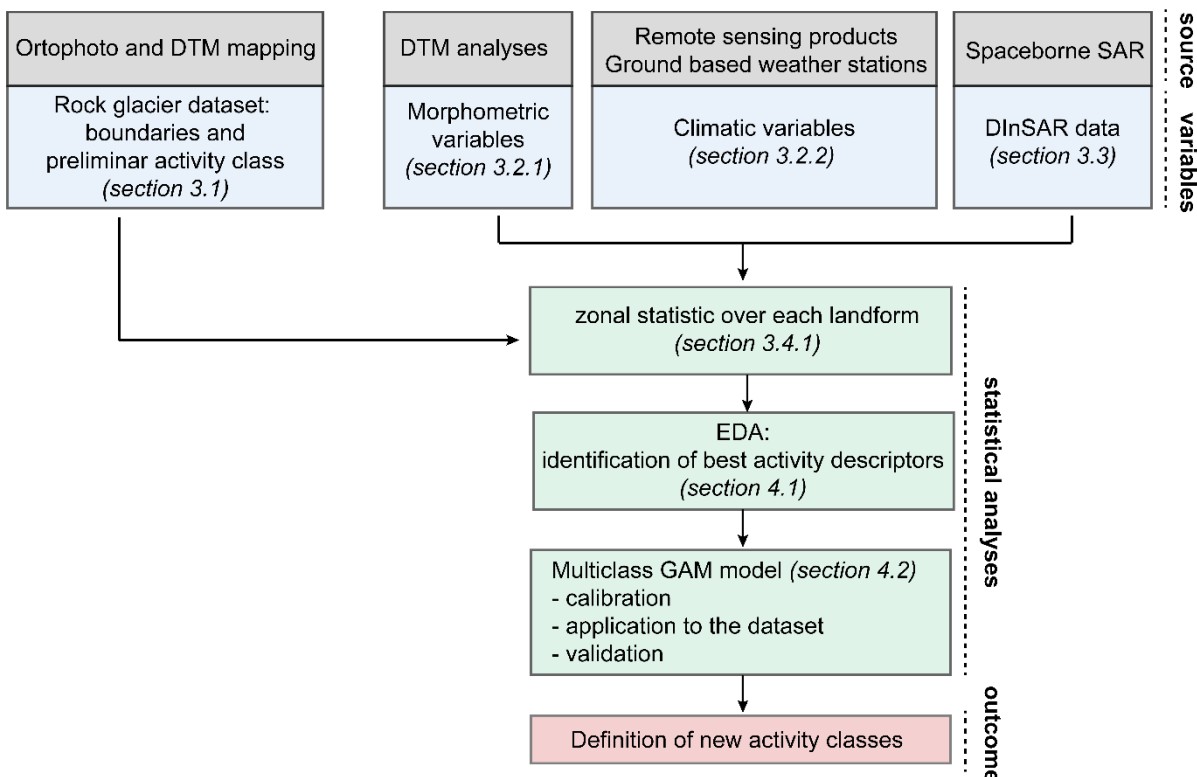

**Figure 2:Schematic workflow illustrating the variables employed and the steps involved in the statistical analysis to derive the final activity class.**

### 3.1 Rock glaciers dataset

This study utilized a comprehensive rock glaciers dataset (yr. 2010) covering the entire South Tyrol region. The
identification and mapping of periglacial landforms were conducted by using LIDAR (Light Detection and Ranging) Digital Terrain Models (DTMs) with a ground sample distance of 2.5 meters, supplemented by orthophotos from 2000, 2006, 2008 and 2014. The resulting catalogue includes not only boundary polygons but also incorporates descriptive features and qualitative aspects, determined through visual morphological inspection of each form. Employing this approach, a classification attribute has been assigned, categorizing forms, where feasible, into active, inactive, and relict states (table1).





This dataset is freely available on the WebGIS portal of the provincial administration of South Tyrol (https://geokatalog.buergernetz.bz.it/geokatalog) and accounts for 1779 features. The 13.5% is active, 70% classified as relict and 3.3% as inactive. The remaining part could not be classified (n.d.) based on a simple geomorphological approach. Starting from the same catalogue, Bertone et al., 2019 reclassified all the features by adopting an interferometric coherence-based approach, that was used as indicator of displacements. Considering only a kinematic approach the features were

reclassified as (i) "moving", for those rock glaciers with displacement detectable using coherence and (ii) "no-moving" rock glaciers where displacement was not detectable (table1). Based on this classification, 13 % of the mapped features is moving, 68% is not moving and the remaining 18% could not be classified simply with the SAR interferometric (InSAR) coherence approach because of vegetation cover, too small dimensions of the rock glaciers or layover and shadowing conditions.

The two classifications offer distinct kinematic attributes, with one (Autonomous Province of Bolzano/Bozen Inventory) focusing on the potential presence of inner permafrost and its morphological expression, while the other (Bertone's Inventory) provides an indication of surface movements. To integrate both perspectives and gather a singular and consistent activity indicator aligned with the newly proposed RGIK classes (RGIK, 2023), we categorized rock glaciers as "active" (A) only if they exhibited movement in both classifications. The "relict" (R) class was assigned to rock glaciers showing no

movement in both datasets and determined to be either inactive or relict. For the remaining cases, which did not fall into the aforementioned categories, we classified them as "transitional" (T), excluding features that could not be classified (table 1).

|  |  | PAB classification (2010): morphological approach | | | |
|---|---|---|---|---|---|
|  |  | **Active** | **Inactive** | **Relict** | **n.d.** |
| Bertone et al., 2019: DInSAR coherence | **Moving** | A | T | T | n.d. |
|  | **Not moving** | T | R | R | n.d. |
|  | **n.d.** | n.d. | n.d. | n.d. | n.d. |

**Table 1: table reporting the activity attributes of the Autonomous Province of Bozen (PAB) classification (row) and those from Bertone coherence-based classification (column). A (= active), T (= transitional), R (= relict) and n.d. (= not defined) correspond to the new preliminary labels attributed combining the two initial attributes.**

This reclassification serves to diminish uncertainty in categorizing the A and R forms, as these groups align more consistently. However, greater uncertainty is associated with the T class, where the two classifications do not converge.





## 3.2 Rock glaciers dataset

To comprehensively characterize the rock glacier area, we extracted terrain attributes linked to local topographic and climatic site conditions, along with area characteristics that can influence debris supply, like the main lithology of the area
(table2).

| Type of variable | Parameter | Unit of measure | Description |
|---|---|---|---|
| Morphometric | Lithology | categorical | Affects the debris supply capability of the catchment and the size of blocks |
| | Total insolation | kWh / m$^2$ | Measure of the solar radiation energy received on a given surface |
| | Slope | ° | Effect on frost- and gravity-driven processes |
| | Aspect | ° | Dip direction of the landform. Controls on the received solar radiation |
| | Elevation | m. a.s.l. | Influence on climatic conditions controlling permafrost distribution |
| | Vector Ruggedness Measure (VRM) | / | Index of surface heterogeneity and harshness |
| | Convergence | / | Outlines channels (convergent) and ridges (divergent) |
| | Profile Curvature | 1/m | Parallel to the direction of maximum slope. Distinguish between concave (negative) and convex (positive) topography |
| Climatic | Land surface temperature (LST) | °C | Radiative skin temperature of the land surface: related to the energy budget of permafrost environments |
| | Precipitations | mm | Liquid and solid precipitations from interpolation of ground weather stations data |
| | Snow cover duration (SCD) | days | Number of days a particular place was covered by snow |

**Table 2: spatial environmental parameters extracted for each rock glacier.**





### 3.2.1 Morphometric variables

The lithology varies significantly across the AOI (Fig.1) due to the juxtaposition of rocks from different geodynamic settings. Starting from a geological map of South Tyrol (scale 1:25000 derived from CARG surveys at 1:10000 scale; http://www.provincia.bz.it/costruire-abitare/edilizia-pubblica/geologia-e-prove-materiali.asp), and based on the lithologies origin (sedimentary, igneous, or metamorphic), we categorized them into four macro classes: i) granitoids and volcanic rocks; ii) metasediments and low metamorphic facies; iii) facies from middle to high metamorphism; iv) sedimentary cover.

Then, as our main goal is on classifying the activity class of mapped rock glaciers rather than analyzing factors contributing to their initiation, we also incorporated morphological indexes sensitive to various permafrost dynamics (table2). Active landforms should in fact display a more swollen appearance due to the presence of inner permafrost and their deformation that often leads to the formation of furrows and transversal ridges inducing a consequent increase in surface roughness. On the other hand, relict rock glaciers, with limited or absent permafrost core, may have a more convex and flatter surface with

consequent lower values of Vector Ruggedness Measure (VRM) and positive profile curvature.

Terrain attributes obtained from a smoothed 10 m DEM resolution (down sampling of the 2006 digital elevation model at 2.5 m from the Autonomous Province of Bozen http://geokatalog.buergernetz.bz.it/geokatalog/#!) were incorporated into the analysis as they are expected to capture the overall characteristics of the topographic niche of rock glaciers. All the analyses have been done in SAGA GIS and ARCGIS 10.8.


### 3.2.2 Climatic variables

*Land surface temperature*

Land surface temperature (LST, Fig.S1) represents the radiative skin temperature of the land surface, as measured in the direction of the remote sensor. While acknowledging the disparities between ground surface temperature (GST) and LST,

this latter generally displays a pattern that may closely follow the GST variability suggesting the possibility of linking GST to LST products (Serban et al., 2023; Sun et al., 2015). As consequence, due to the lack of ground measurements that could be used to retrieve the GST, we considered a Landsat derived LST as indicator of surface temperature variability, sensitive to factors such as elevation, slope, aspect, soil structure, snow, and vegetation cover.

Analyses were carried out on Google Earth Engine (GEE) platform by using the code proposed by Ermida et al. (2020) to

process thermal infrared (TIR) band signals provided by Landsat 8 over the period 2013 to 2023 (table3).

| Satellite | Bands | Wavelength (μm) | Dataset | ground resolution | time period |
|---|---|---|---|---|---|
| Landsat 8 (OLI; TIRS) | Red: B4 | 0.64–0.67 | C01/T1_SR | 30 m | Septembers from 2013 |
| | NIR: B5 | 0.85–0.88 | C01/T1_SR | 30 m | |
| | TIR: B10 | 10.6–11.19 | C01/T1_TOA | 100 m * | to 2023 |

**Table 3:dataset used to compute land surface temperature (LST) in Google Earth Engine (GEE).** * Resampled to 30m



The distribution of the rock glaciers spread across a wide range of elevation (1850-3100 m a.s.l.). Since several of them, especially those classified as R, occupy low altitude sectors and can be covered by bushes and shrubs, some precautions were imposed on the algorithm to quantify the LST properly such as the correction of NDVI (Normalized Difference Vegetation Index) to emissivity to adjust it for the surface vegetation contribution (Malakar et al., 2018, Parastatidis et al., 2017, Ermida et al., 2020). A cloud filter was also added to exclude images with a cloud coverage exceeding 20% over the scene. For the analyses, we then only consider images acquired in September to emphasize potential spatial differences between rock glaciers bearing permafrost and areas with no permafrost. In September, as the air temperature begins to drop, the differing response rates of permafrost and rocks to this change can lead to a more pronounced temperature delta. In addition, after the warmer summer months, permafrost may still be in the process of thawing during which heat absorption phenomena occurs, contributing to a slower increase in temperature compared to rocks without permafrost.

*Precipitations*

Mean seasonal precipitation maps (Fig.S2, S3) were extracted starting from high-resolution gridded datasets (cell size of 250 m) of daily precipitation records for Trentino South Tyrol (Crespi et al., 2021).

We analyzed a twenty-year timeframe spanning from 2000 to 2018, calculating the average precipitation values for both summer (July to September) and winter (from October to June) seasons. This differentiation is crucial in high-altitude environments due to the necessity of discerning between periods dominated by liquid precipitation in summer and those characterized by solid precipitation in winter months. This is particularly important because the weather station in South Tyrol collects precipitation data without distinguishing between these two contributions.

*Snow Cover Duration*

Snow cover duration (SCD, Fig.S4) on the ground significantly affects the ground thermal regime modifying the heat insulation, water storage and runoff contribution, but the interaction of ground temperature and snow cover is not entirely straightforward and its effects on permafrost conditions can change according to snow depth, type, and water content (meaning snow water equivalent, SWE; Bender et al., 2020). However, the analysis of the thickness and SWE of the snow cover goes behind the scope of this study and would require additional information from ground measurements. Here, we only consider the snow cover duration, retrieved from MODIS satellite with 250 m spatial resolution, as number of days per year having a multispectral indication of snow on the ground (Notarnicola et al., 2013). Using this SCD parameter, a potential correlation between the rock glaciers' activity at a regional level was made. In fact, SCD was used not only to investigate into the predisposing factors that lead to the formation/absence of rock glaciers but also to understand how the temporal duration of snow cover might relate to the observed activity of the rock glaciers in the specific AOI.



## 3.3 InSAR data

We utilized Copernicus Sentinel-1 C-band single-look complex (SLC) radar data acquired in the snow free period between July and October spanning the years 2020 to 2022. The combination of Sentinel-1's extensive swath width and rapid revisit time renders it well-suited for monitoring widespread landscape-scale deformations. The data were collected in interferometric wide (IW) swath mode with a swath width of 250 km, employing vertical co-polarization (VV) along both ascending orbits 117 (68 SAR images-165 interferograms), as well as descending orbit 168 (57 SAR images-133 interferograms). The Sentinel-1 SLCs exhibit a spatial resolution of 22 m in the azimuth (along-track direction) and 2.7-3.5 m in the range (across-track direction).

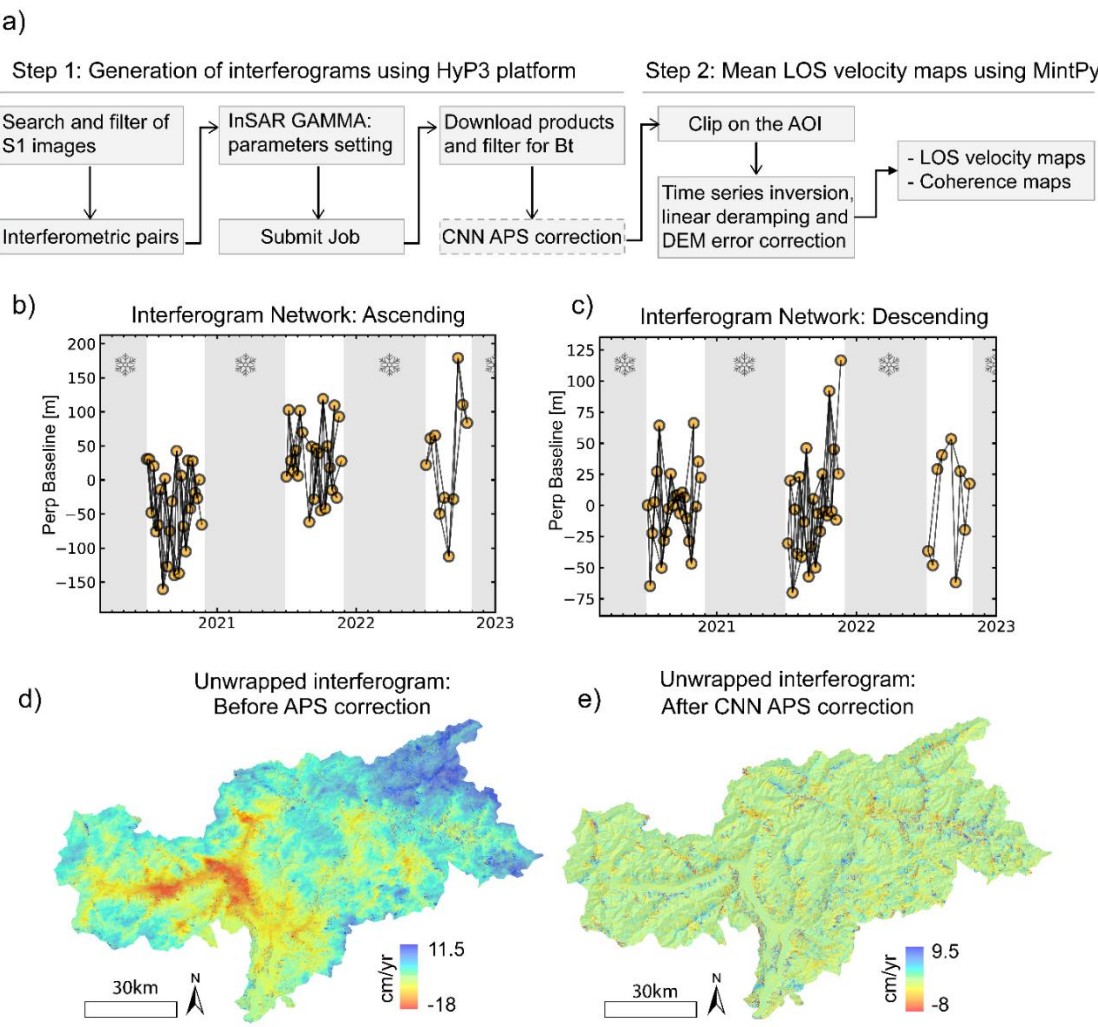

**Figure 3: First and second steps of the DInSAR processing chain: a) workflow implemented in HyP3-MintPy and CNN correction; interferometric pairs elaborated for the ascending (b) and descending (c) geometry; d) example of an unwrapped interferogram without CNN APS filtering and with CNN APS filter (e).**



We set a revisit time range of 6, 12, 24 and 30 days and computed interferometric pairs employing the Small Baseline Subset (SBAS) processing of Sentinel-1 data through Alaska Satellite Facility's Hybrid Pluggable Processing Pipeline (ASF HyP3, Fig 3), a web-based SAR data processing platform that primarily utilizes Amazon services. Multi-looking was performed, involving 10 looks in range and 2 looks in azimuth, resulting in interferograms with a pixel spacing of about 40 m. ASF HyP3 then utilized the 2021 release of the 30 m Copernicus "GLO-30" digital elevation model (DEM) product to eliminate

the topographic component of the phase and geocode the interferograms. The Atmospheric Phase Screen (APS) contribution in interferograms was filtered through a convolutional neural network (CNN) approach (Brencher et al.,2023). This method was employed to eliminate both stratified and turbulent atmospheric noise.

The key strengths of this approach stem from its independence of external atmospheric data or synthetic training data; instead, corrections are derived directly from the observed ones. After the atmospheric filtering, unwrapped interferogram

were re-ingested in Python-based Miami InSAR Time-series software MintPy (Yunjun et al., 2019) to produce mean Line-of-Sight (LOS) displacement rate maps. MintPy works based on a weighted least squares inversion formula (Augustan et al., 2022, Yunjun et al., 2019). By default, and starting from the interferometric stack, it estimates the average velocity as the slope of the best fitting line to the displacement time-series corrected for the APS contribution. All deformations are referred to a single point within the analysis that is automatically selected among the pixels with high average spatial coherence

(>0.85, Yunjun et al., 2019).

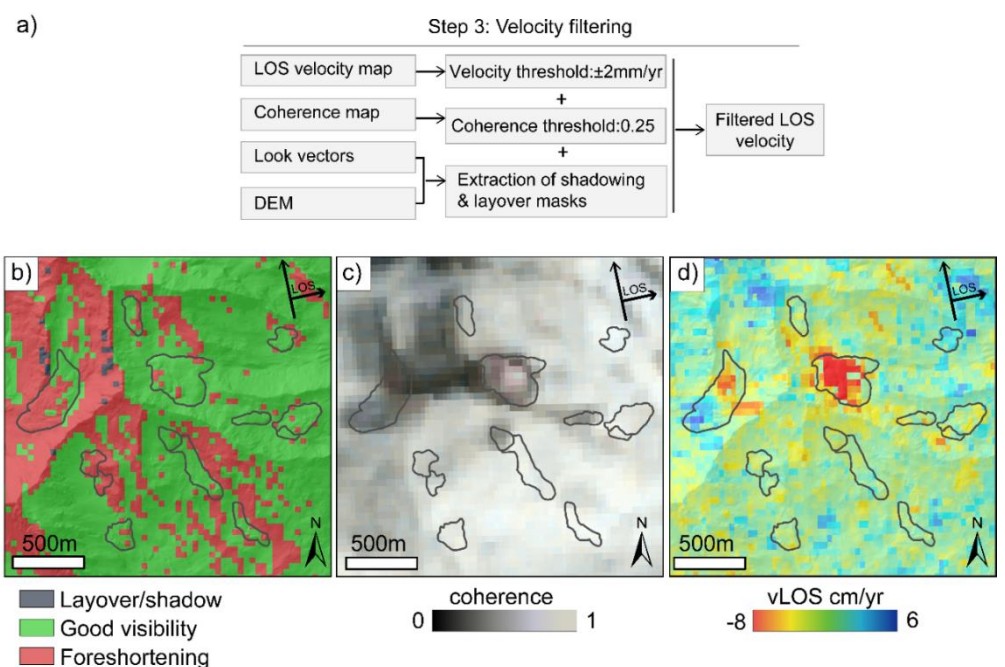

**Figure 4: Third step of the DInSAR processing chain: a) workflow adopted to filter velocity product applying coherence, velocity, and topographic masks; b) visibility map; c) coherence map; d) final filtered velocity map.**



For each polygon we selected the most suitable acquisition geometry, and we provided the corresponding C index (Notti et al. 2014) to indicate how well each landform is caught from the satellite according to the combination of slope, aspect and satellite orbit parameters (LOS, orbital and azimuth angle). In refining the velocity maps, we employed coherence, velocity, and topographic filters (Fig.4) to remove pixels with high uncertainty due to geometric and displacement uncertainties. Areas affected by layover and shadowing were discarded from the displacement map, as well as areas with coherence values under 0.25.

**3.4 Data extraction and integration**

3.4.1 Extraction of environmental and climatic statistics

For each rock glacier polygon, mean values for environmental and climatic variables were assigned based on the values within the polygon boundary. Furthermore, for DInSAR-related variables (i.e., velocity and coherence), additional statistical descriptors were extracted such as variance, 25th-75th, and 90th percentiles. Finally, each rock glacier was given an estimation based on how much of its area is covered by the filtered SAR data (Fig.4). This information can be considered a measure of uncertainty associated to the data based on the spatial coverage within each polygon.

Starting from the distribution map of the rock glaciers and considering their displacement range, we then made two distinctions: (i) movements strictly related to periglacial processes that are confined within mapped rock glacier boundaries and (ii) movements less influenced by permafrost creep and deformation mechanisms and lacking respective morphological evidence that are placed in the surrounding areas of polygon boundaries. To accomplish this diversification, around each mapped landform, at a distance of 100 m, a 100 m wide buffers (Fig.S5) was generated to address areas with no visible displacement ascribable to periglacial deformations. In the case of adjacent forms or multiple rock glaciers coalescing into one body, these rims were cut to avoid overlaps between different features.

The delineation of surrounding areas external to the rock glaciers serves a dual purpose: facilitation in the comparison step between parameters measured inside the periglacial landform and its immediate surroundings and secondly it permits the differentiation of contributions from permafrost movement and potential deformations (such as gravitational movements) which could affect the slope stability. Consequently, we also computed the delta of values between the interior and exterior of the rock glaciers. This calculation accentuates variations (e.g. velocity difference) that may be attributed to the presence and activity of permafrost or other sources of deformations.

3.4.2 Statistical modelling

To discern the key factors influencing the distinction between A, R, and T rock glacier classes, we performed an initial Exploratory Data Analysis. This exploration served to inform the selection of explanatory variables by assessing their potential impact on defining the activity class and examining their relationships with the response variables. Subsequently, a GAM was employed to investigate the associations between the chosen predictor variables derived from both environmental and DInSAR datasets and the response variables.





GAM provides a versatile framework for examining non-linear associations between the response variables (here, the three activity classes of rock glaciers: A, T and R) and continuous variables (e.g., morphometric and DInSAR indexes) by enabling the incorporation of both parametric and non-parametric covariates, facilitating the exploration of individual effects (Brenning, 2010). The initial phase of the model construction involved the determination of smoothing parameters, which control the flexibility of the model, for continuous variables. This process utilized internal cross-validation, with a constraint of four effective degrees of freedom for spline parameterization. The significance of each term was assessed based on p-values, with the null hypothesis (no effect of the term) being rejected at a threshold of 0.05. Consequently, only terms demonstrating a significant effect (p-values < 0.05) were incorporated into the final model (Wood, 2013). In the model set up, we examined not only the individual predictors influence outcomes but also considered interaction terms. Interactions can in fact reveal relationships that may not be apparent when considering single predictors. For example, understanding how morphometric characteristics and DInSAR indexes interact can help to uncover the mechanisms driving the activity classes of rock glaciers. Additionally, we utilize the Accumulated Local Effects (ALE; Apley and Zhu 2020) approach for GAM to interpret the influence of each predictor variable on the model, providing insights into their respective impacts on the response variable. The GAM model was then fitted to the data and its performance evaluated using receiver operating curve (ROC) analysis for a multinomial response variable with three classes. Once verified its performance, the model was finally applied for the classification of unknown features, and predictive performance estimates were computed through multiple independent test sets employing 2, 5, and 10 k-fold cross-validation selections. The calculation of predictive performance involved iteratively dividing the original dataset into training data (utilized for model fitting), and test data (employed for calculating the AUROC metric).

## 4 Results

### 4.1 Exploratory Data Analysis

The exploratory data analysis allowed first insights into empirical associations between the rock glaciers classes and potential predictor variables.

Considering the classification in Table 1, we analyzed the distribution of morphometric and environmental variables among the A, T and R classes. In this step, boxplot and conditional density plots were used to highlight the distribution of continuous variables over the three classes, specifically focusing on parameters where the interquartile ranges display higher divergence among classes. In this analysis, both rock glaciers related values (computed inside the boundary of the landforms), and delta values (computed as difference between rock glacier and the corresponding outer area), were considered. In particular, the variables that resulted most significative are LST (land surface temperature), SCD (snow cover duration), VRM (vector ruggedness measure), mean and variance values of coherence, coherence difference (from inside the rock glacier and the area outside), variance in velocity and highest velocity (Fig.5). Land surface temperature (Fig.5a) serves as a discerning factor delineating among distinct categories of landforms: R, T, and A rock glaciers. R landforms exhibit the



highest temperature regime, characterized by a mean LST value of approximately 17°C. T landforms occupy an intermediary

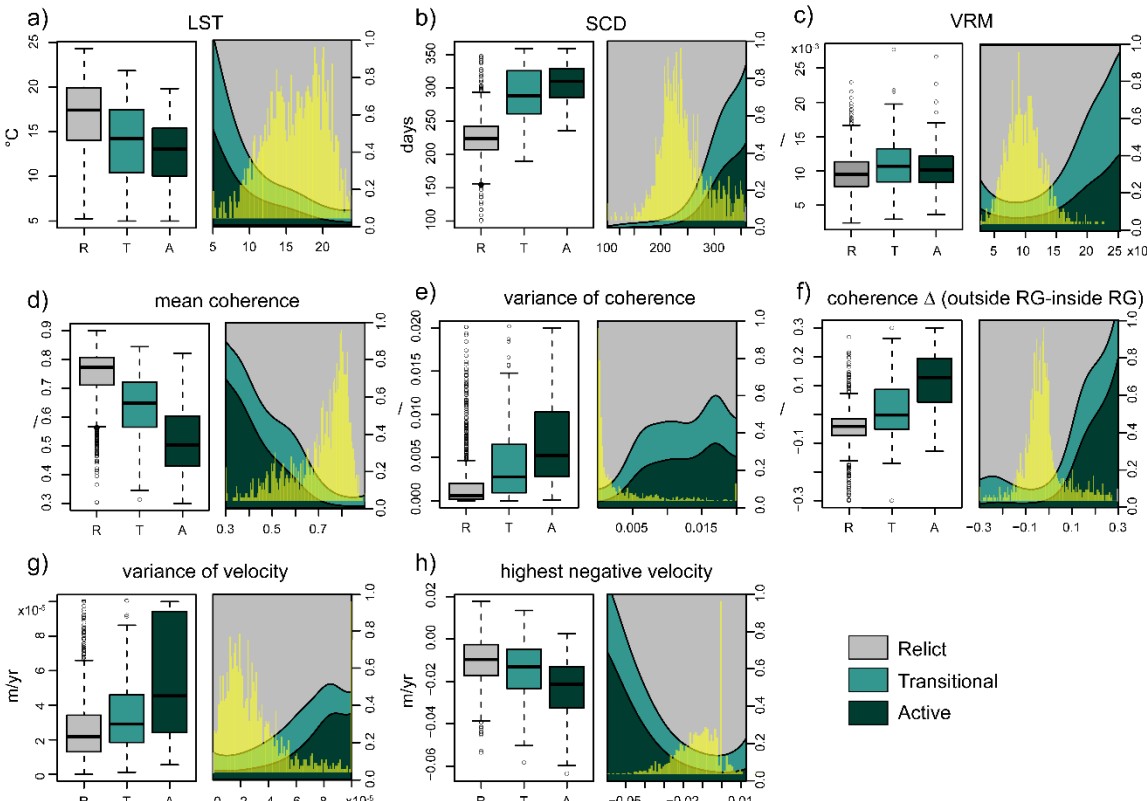

**Figure 5: variables retained for the discrimination of the three activity classes of rock glaciers. Both boxplots and conditional density plots are reported for each variable. Boxplots shows the distribution of values for each variables in each class, conditional density plots describe how the conditional distribution (0-1) of the categorical variables A-T-R and y changes over a numerical variable; a) land surface temperature-LST; b) snow cover duration- SCD; c) vector ruggedness measure- VRM; d) mean coherence; e) variance of coherence representing the variability of coherence values inside rock glacier polygons; f) coherence Δ, computed as difference between the mean values inside the 100m rim of each rock glacier and mean value inside the rock glacier; g) variance of velocity; h) highest velocity values. Negative values are considered to retain only movements away from the satellite.**

position, displaying a mean LST value of around 14°C. Meanwhile, A rock glaciers share a similar mean LST value to T ones, albeit with lower maximum temperatures. This differential temperature ranges across the rock glacier classes underscores the utility of LST as a diagnostic parameter for delineating the thermal conditions favouring or limiting the

activity of these landforms. Similarly, SCD exhibits notable discriminative characteristics among the various rock glacier classes. R rock glaciers demonstrate a mean SCD of 225 days, whereas T and A classes display longer durations, with respective values of 290 and 310 days of snow cover (Fig.5b). Here SCD solely reflects the presence of snow cover on the ground without providing details regarding snow depth or water content within the snowpack. Surface roughness (Fig.5c), expressed by VRM, also provides an indication of the surface conditions controlled by permafrost deformation, with T and A

classes holding a slightly higher VRM than R ones. Therefore, recognizing its significance in representing potential surface



variations, we retained this parameter and incorporated it into the subsequent analysis in conjunction with other parameters. Coherence-related metrics such as mean coherence value (Fig.5d), coherence variance (Fig.5e), within each polygon, and coherence delta (Fig.5f), between the rock glacier landform and the surrounding 100 m rim emerge as highly discriminative indicators among the three activity classes, as evidenced by boxplots that exhibit minimal overlap. Generally, A rock glaciers

have a low coherence value, mean value around 0.5, while R rock glaciers, which keep a higher surface stability, reach values of 0.8. Velocity values prove to be an effective classificatory too, especially considering vLOS variance (Fig.5g), related to internal surface variations of velocity, possibly between discrete sectors or lobes, and the highest velocity value recorded in each rock glacier (Fig.5h). Less influence is exerted by other morphometric indexes (Fig.6) like as aspect, slope, total insolation, curvature, and convergence that did not lead to an evident distinction among the classes. In contrary,

elevation shows a high capacity in separating R class from T and A classes. However, we did not consider this variable since its contribution also affects the LST measure that has a strong elevation related trend. Keeping the elevation parameter would have added redundancy in data.

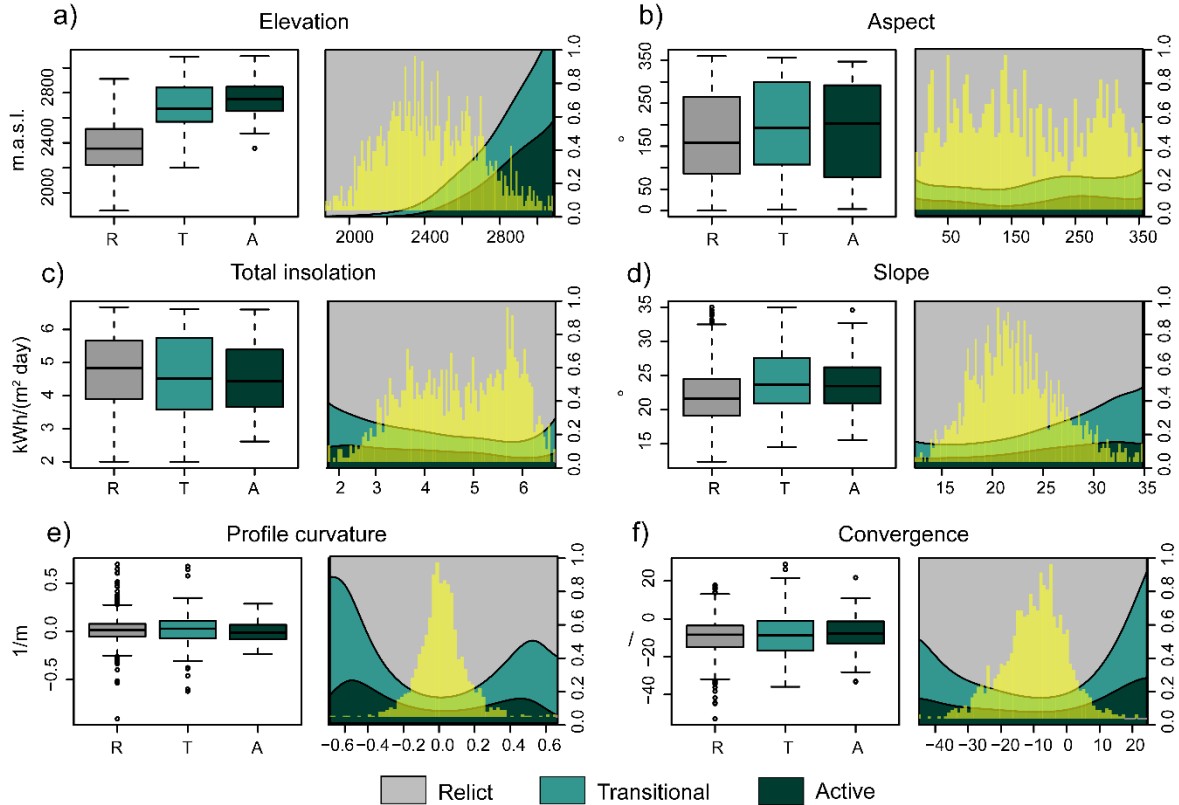

**Figure 6: morphometric variables that were not retained in the analysis due to their little discriminant capacity between activity**
**classes: a) elevation; b) aspect, computed as mean direction towards north; c) total insolation coming from the combined contribution of direct and diffuse insolation; d) slope angles; e) profile curvature: positive values indicate concave structure of the landform, negative values convex shapes; f) convergence: positive values indicate divergent areas, negative values convergent areas.**





## 4.2 Multiclass GAM model

After the above steps, we considered the eight selected variables (Fig.5) as predictor variables in a multinomial response variable GAM model which included smooth terms for specific variables and tensor product smooth interactions between pairs of variables, all using thin-plate regression splines with a smoothness parameter of 4. The decision to incorporate tensor product interactions, specifically between variables such as mean coherence and coherence Δ, as well as variance of coherence and variance of velocity, was driven by considering that diverse values of coherence might also be reflected in

higher variations of coherence between the inside and outside of the landform. Similarly, variations in variance of coherence may be associated with variations in variance of vLOS as consequence of increased surface deformations and terrain alterations. These interactions aim to capture the potential interplay and mutual influence between these variables, acknowledging that their joint effects on the response variable may not be adequately captured by single smooth terms.

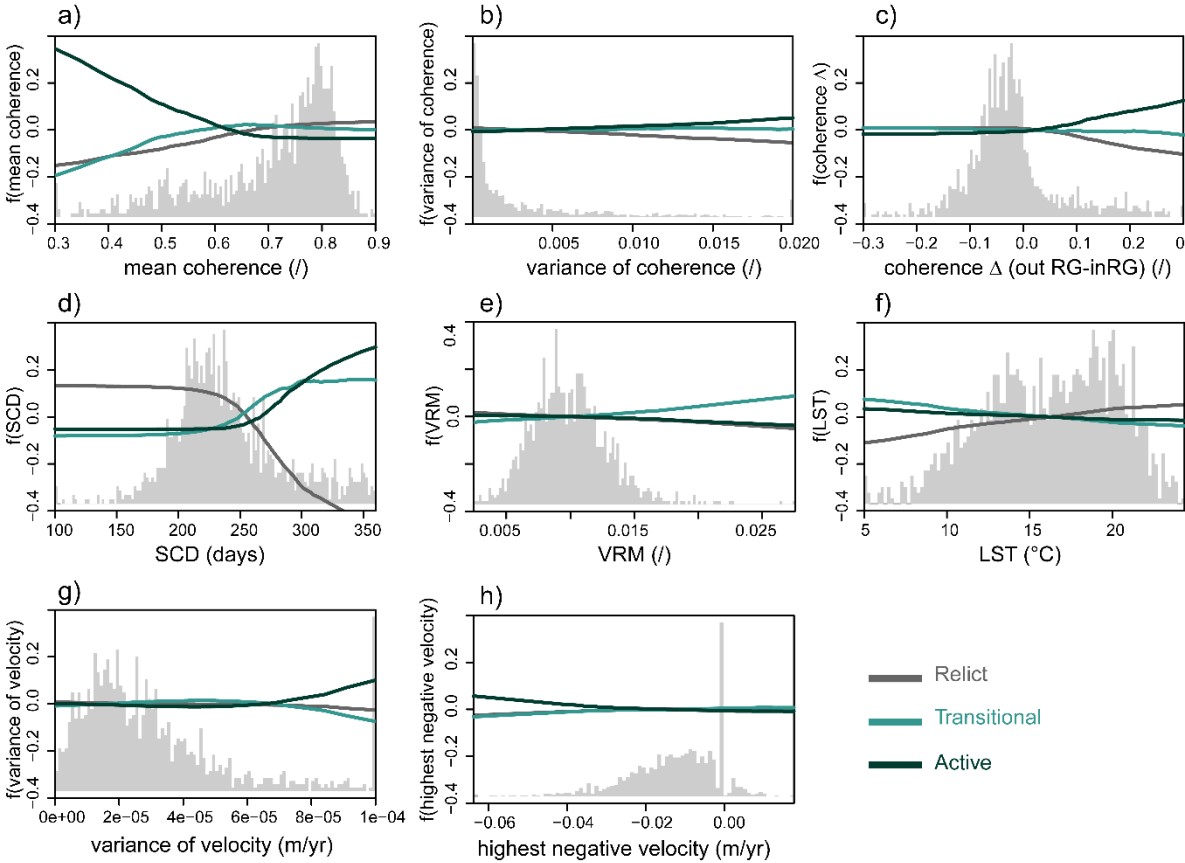

**Figure 7: ALE plot for predictor variables. Each plot shows how the predicted outcome varies with changes in individual predictor variables, while holding other variables constant. Movement of the line indicates shifts in predicted probability for a class as the predictor variable changes: steeper slopes denote greater influence of the predictor variable on the predicted probability. x axis represents the range of values of each variable, the y axis refers to the main effect of the feature compared to the average prediction of the data.**





In addition, considering the interaction between these terms also led to an improvement in model performance, as indicated

by lower Akaike Information Criterion (AIC; Akaike, 1974) values (AIC=1264 considering tensor product interactions, AIC=1271 considering single smooth terms).

As feature effect method we adopted an accumulated local effect (ALE; Apley and Zhu, 2020) representation to inspect the effect of changes in the value of each predictor variable on the model's predictions.

The value of the ALE can be interpreted as the main effect of the feature at a certain value compared to the average

prediction of the data. In Fig.7, it's evident that the mean coherence and the SCD values (Fig.7a and 7d, respectively) are the major control factors in distinguish the T, A, R classes. For the A class, the ALE plot exhibited a descending trend as coherence increased. Notably, higher coherence values are associated with lower predicted responses, suggesting a negative impact or a diminishing effect on outcomes within the A class as coherence increased. On the contrary, R and T classes initially displayed negative y-values for low coherence, indicating that lower coherence is associated with lower predicted

responses. However, as coherence increased, the effect transitioned to positive y-values, resulting in an overall ascending trend. This observed pattern suggests that higher coherence values are associated with higher predicted responses for the T and R classes. An opposite trend is found for SCD (Fig.7d) which shows how, over a certain number of days, there is a positive effect of snow cover duration on A and T rock glaciers, while R are influenced negatively. Slight differences in the LST (Fig.7f) and velocity which can also be correlated to a different capability in predicting the activity classes. LST values

around 5°C are more representative of conditions proper of A and T rock glaciers. This is probably due to the presence of internal permafrost and the occupied topographic area (Fig.6a). Higher temperatures on the contrary correspond to an increase in predictability of R classes.

## 4.3 Fitting performance evaluation and model extension

We used the Receiver Operating Characteristic (ROC) and in particular the area under the curve (AUC) metric to evaluate

the performance of our classification model across different thresholds. In the case of multiclass classification, a notion of TPR (True Positive Rate) and FPR (False Positive Rate) is obtained after binarizing the output. This can be done according two different schemes: i) one-vs-rest scheme, which compares each class against all the others (assumed as one); ii) one-vs-one scheme, which compares every unique pairwise combination of classes.

The evaluation of our multi-class classification model yielded an AUC of 0.87 in the One-vs-One (OvO) scenario and 0.95

in the One-vs-Rest (OvR) scenario (Fig.8). These AUC values signify a strong overall performance in distinguishing between the three rock glacier classes, further supporting the effectiveness of the GAM in capturing the relationships within the data. The lower AUC value for T class vs. A class and T vs R classes might indicate that the model faces challenges in discriminating between these classes, possibly due to the disparity in class frequencies (higher number of relict forms vs active and transitional ones) or a higher uncertainty associated with the identification of T landforms. These AUC values

signify the model's consistent ability to distinguish between individual classes when compared to the rest and the effectiveness of discriminating one class against the collective set of other classes. The metrics, surpassing the 0.5 baseline,



underscore the model's efficacy in capturing relationships within the dataset, suggesting its potential utility for accurate classification across diverse categories.

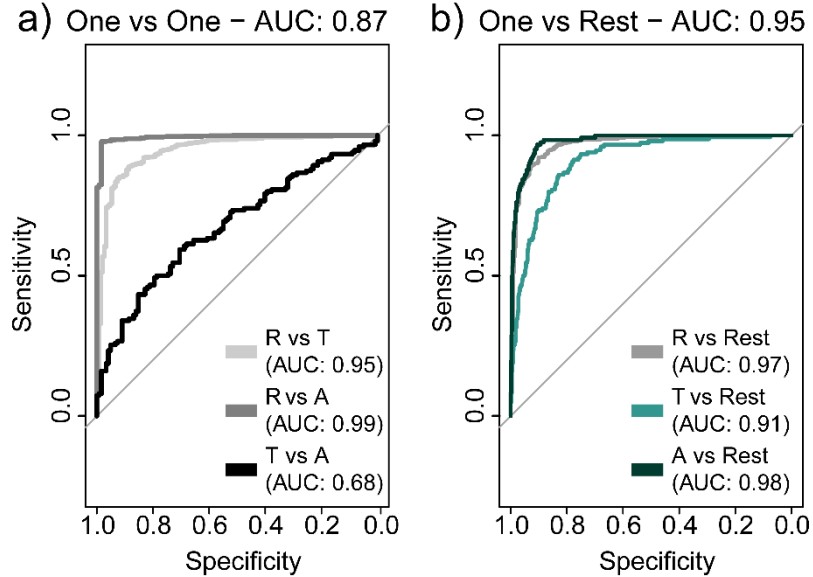

**Figure 8: multiclass model performance evaluated for each class as a) One vs One and b) One vs Rest.**

Once evaluated the predictivity capability of the model, we applied it to the entire regional dataset, considering also the n.d.
landforms (table1) to predict the most probable class. The model's performance was evaluated by assessing the match between predicted and true classes, as well as measuring the proximity to the nearest class in terms of probability.

The robustness and discriminative performance of the classification model were assessed through repeated k-fold cross-validation (Fig.9a). The cross-validation approach, employing 2-fold, 3-fold, 5-fold, and 10-fold splits, was implemented to systematically evaluate the model's generalization across various train-to-test ratios. The performance of the model was
quantified using the AUROC. The resulting boxplot visually depicts the distribution of AUROCs across different cross-validation scenarios, offering insights into the model's stability and discriminative prowess.

1716 rock glaciers over 1779 were classified and 63 could not be classified due to the lack of data, such as the invalid velocity pixels which were excluded for coherence or topographic effects. The spine plot in Fig.9b illustrates the correspondence between predicted and initial classes, with each spine representing a predicted class and the height of its
385 segments indicating the proportion of observations assigned to each initial class within that prediction. The conditional density plot (Fig.9c, 9d) further delves into model behaviour by showcasing the distribution of predicted classes across varying degrees of uncertainty, depicted along the x-axis as the uncertainty index. Fig.9c depicts the distribution of uncertainty in the complete dataset, while Fig.9d only represents the uncertainty distribution in the n.d. cases. Values close to 1 points out a higher confidence level, while lower values indicate a higher uncertainty in the classification. Being the most





abundant ones, the R features, are characterized by a high confidence level, whereas this level decreases considerably for the T class which show the highest uncertainty in the prediction.

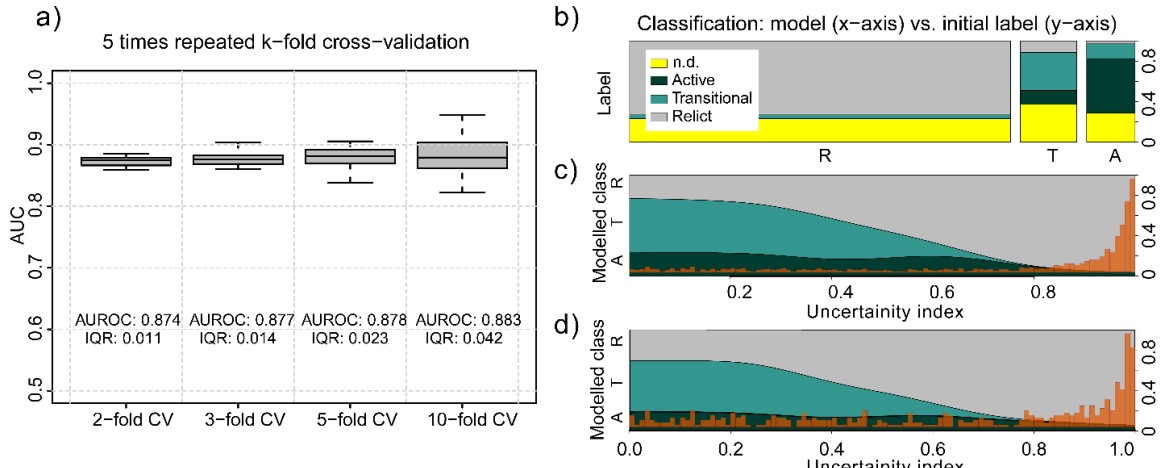

**Figure 9: Model performance and fractional breakdown of the obtained activity classes: a) boxplot showcasing the AUC metric for each k-fold cross-validation.; b) spinogram illustrating the distribution of classified rock glaciers based on both our model and the initial classification.; c) conditional density plot for uncertainty values over all the observations; d) conditional density plot of uncertainty values only over not determined (n.d.) features. Red histograms indicate the distribution of uncertainty index values for the plotted observations (all the observations in Fig.9c, only n.d. observations in Fig. 9d).**

Upon reclassification, approximately 67% of the initially proposed classifications remained unchanged, i.e. rock glaciers classification fitted Bertone and PAB label as reported in Table 1. Conversely, approximately 32% of the landforms were reclassified into different categories. The spatial distribution of newly attributed activity classes and the agreement/not agreement among the initial classification label is shown in Fig.10a, whereas in Fig.10b the regional distribution of the rock glaciers, adopting the new classification, is shown. Respect to other methods, our model additionally offers estimations of predicted probabilities for each class (Fig.10c), with relict forms exhibiting the highest level of confidence (indicated with different border colour lines in the figure). This heightened confidence of the R (Fig.10d) respects the other two categories is partly attributed to the great diversity in number of the rock glaciers into each class. The R class shows the highest level of confidence (> 0.8) probably due to the greater abundance of them (1345) respect to T and A. The A class (formed by 171 rock glaciers) shows an intermediate confidence interval in our dataset whereas the T (formed by 200 features) exhibits the lowest confidence level (< 0.4), primarily due to the inherent mismatch between their geomorphological parameters and coherence-based attributes, as evident in Table 1.





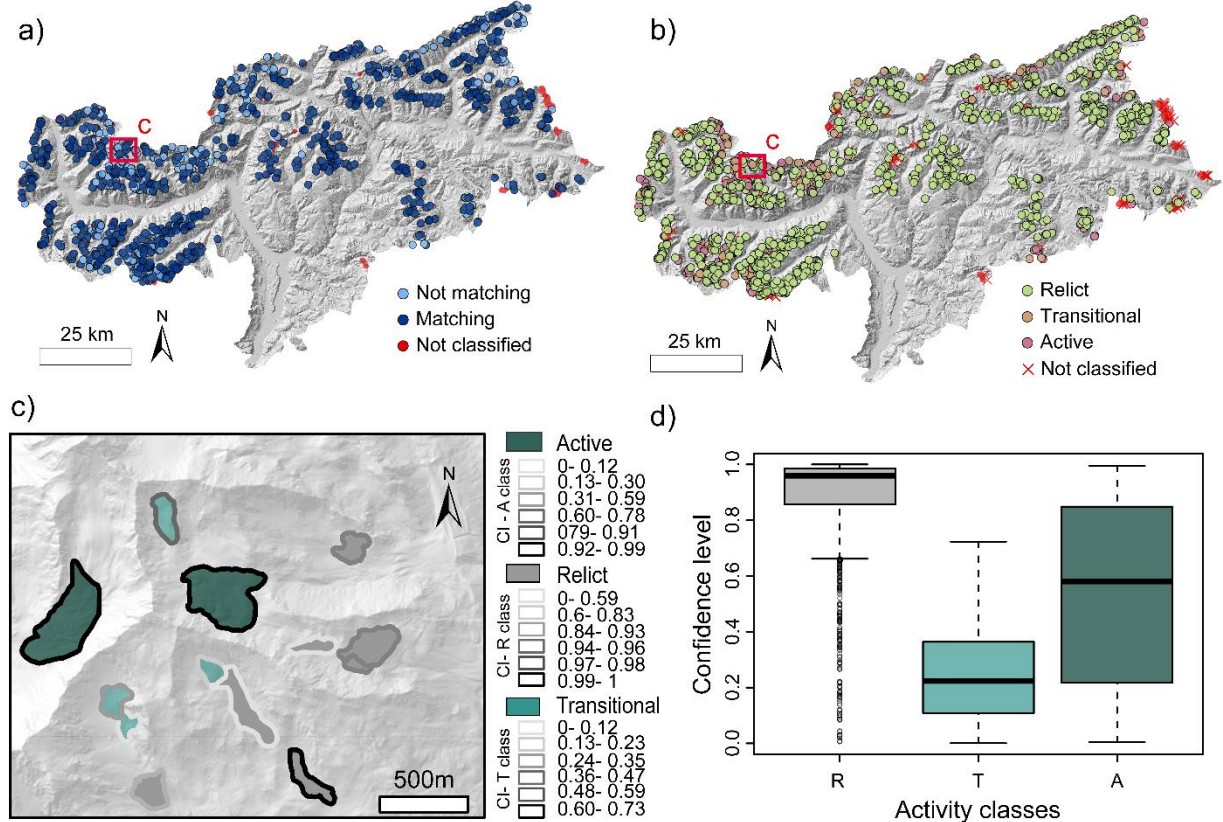

**Figure 10: model classification outputs: a) distinction between matching and not matching activity class between the initial label and the new one; b) new rock glaciers classes; c) example of the attributes associated to some rock glaciers in the area identified within the red square. The colours of the borders correspond to the confidence interval (CI) for each feature in the activity classes. Values close to 1 point out a higher confidence, lower values higher uncertainty in the prediction; d) distribution of classification uncertainty in each class.**

## 5 Discussions

### 5.1 Classification approaches

The growing interest in periglacial landforms, particularly rock glaciers within alpine mountain systems, emphasizes the importance of understanding their dynamics in the context of climate change and its far-reaching implications. As climate change effects intensify, with temperature fluctuations and alterations in precipitation patterns, comprehensively quantification of the activity and deformation of these landforms becomes increasingly crucial since it provides knowledge of ongoing changes in the high mountain cryosphere (Kääb et al., 2007).

Various approaches exist for assessing the activity of landforms at a broad scale. These include: i) a morphological method, which involves visually inspecting orthophotos, satellite images, or conducting field surveys to identify diagnostic features



associated with permafrost deformation, such as furrows, ridges, steep fronts, and lateral margins (Scotti et al., 2013); ii) an
interferometric coherence method, as utilized by Bertone et al., 2019, which relies solely on kinematic analysis to differentiate between moving and non-moving landforms based on coherence values; and iii) a velocity method, typically derived from DInSAR data, particularly for regional scale investigations (Kääb et al., 2021, Strozzi et al., 2020; Zhang et al.,2021). While each of these approaches has demonstrated effectiveness in defining the activity state of rock glaciers, they also possess significant limitations when considered alone. The accuracy of geomorphic-based classification is heavily
dependent on image quality and operator expertise, leading to subjective mapping outcomes. Conversely, InSAR-based methods encounter intrinsic limitations inherent to the technique itself, particularly evident in complex environments like the high alpine terrain.

Commonly these techniques aim at validating the results of one method with evidence of another (Bertone et al., 2024; Ma et al., 2024), for instance visually inspecting the presence of morphostructures and displacement related features with DInSAR
surface patterns (Agliardi et al., 2024).

In our study, we just not simply compare the results gathered from the interferometric approach and morphological and climatic ones, but we jointly exploit their descriptive potential to develop a comprehensive statistical model for categorizing mapped landforms into the three activity classes proposed by RGIK 2023: active (A), transitional (T), and relict (R).

We processed both geomorphological and climatic maps (table2), incorporating data from in situ measurements obtained
from weather stations, as well as remote sensing products such as MODIS and Landsat. Through exploratory data analysis, we then selected variables that proved to have a higher discriminatory power in classifying rock glaciers across the three activity classes.

In delineating the activity of rock glaciers, we found that three variables, namely snow cover duration (SCD), vector roughness measure (VRM), and land surface temperature (LST), hold greater significance, with higher quartile distinction
between the boxplots of each activity class or with p values <0.1 as smooth terms in the GAM, compared to traditional topographic factors like slope, aspect, and curvature.

SCD, for instance, plays a crucial role in regulating the energy balance of the land surface, thereby directly influencing melting and refreezing rates within rock glaciers, and thus also controlling the displacement patterns. This result is also supported by previous studies which highlighted the relevance of the snow cover in determining permafrost occurrence at a
local scale (Apaloo et al., 2012), and at the regional scale (Marcer et al., 2017), influencing rock glacier activity distribution by altering the ground thermal regime. Similarly, the VRM, associated to velocity variations, offers valuable insights into surface roughness variations, which directly reflect the flow dynamics within rock glaciers. These two variations manifest as the formation of furrows and ridges, resulting from compressive and tensile stresses associated with different flow velocities and internal deformation interactions with the topography. Additionally, LST serves as a key indicator of heat exchange
processes, offering valuable information on areas potentially hosting permafrost. Despite not being a direct measure of in situ land surface temperature, LST from Landsat proves to be reliable in studying the spatial variability of surface temperature in complex topography (Gök et al., 2024). Here, its application to the periglacial environment is effective in





discriminating areas with lower temperatures influenced by a combination of factors such as, for example, altitude, exposition and ground conditions which can consequently be correlated to permafrost conditions. Using LST as descriptive
variable is thus acceptable even if it does not directly refer to the deeper ground surface temperature.

Although not immediately evident, the relatively minor influence of the other morphometric indexes (i.e., slope, aspect, curvature), likely stems from their primary role as predisposing factors to the initiation of rock glaciers within the study area, rather than exerting significant control over their ongoing activity.

For instance, slope should play an important role in controlling surface velocities which can be described through a creep
law by the joint interaction of slope angle and rock glacier thickness (Cicoira et al., 2020. Kaab et al., 2023). However, our findings do not outline such a clear dependency between velocity and steepness as also reported in Buchelet et al. (2023). Also considering aspect alone we could not find meaningful links with the activity rate. Bertone et al. (2024) got similar results over a sub portion of our same area of study, thus confirming that using aspect as topographic proxy for inferring the permafrost content, and the activity class, may be problematic.

Therefore, to establish the true impact of changes in these variables on activity classes, local scale detailed analyses should be conducted. It is crucial to explore their local influence in site-specific cases, as local conditions (such as lithology, permafrost distribution, and local changes in slope) can significantly influence the activity of rock glaciers. These aspects may modulate factors like ice content, ground temperature, and frictional behaviour, thereby shaping the dynamics of rock glacier movement and activity patterns.

An additional consideration should be given to the precipitation values, which did not display a clear correlation between mean summer and winter values and activity classes at the regional scale. Despite precipitation events are likely contributors to short-term and seasonal variations in the velocity of rock glaciers (Kenner and Magnusson, 2017; Kenner et al., 2021), when analysed at a broader regional scale further investigation is required to catch the quantitative relationships between their class of activity and precipitation levels (Zhang et al., 2023). This is due to local factors that may exert a more
significant influence on controlling rock glaciers activity than broader precipitation patterns alone. Precipitation cannot be considered a singular influencing factor as it strongly interacts with other local conditions (temperature, exposition etc.) in regulating the activity and evolution of periglacial features.



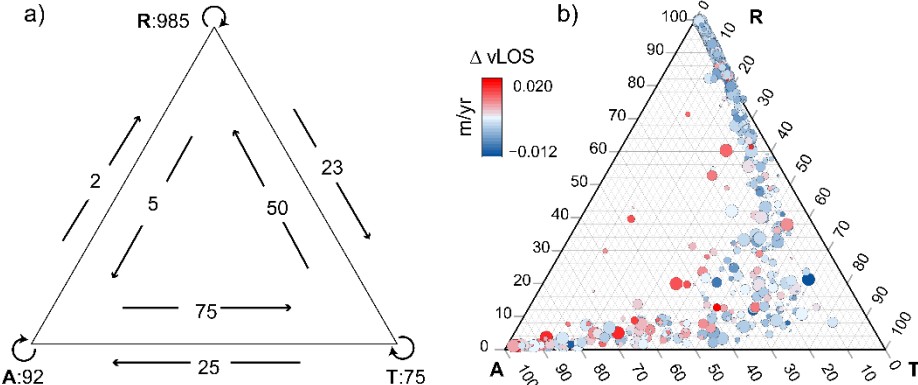

**Figure 11: Distribution of rock glaciers according to the activity class; a) number of rock glaciers retaining their original class after reclassification (vertices of the triangle) and reclassified into different categories (arrows on the sides); b) distribution of reclassified data based on their probability of belonging to each class. The size of the circles is related to the spatial cover of SAE data over each landform. Higher coverage reflects in larger circles and vice versa.**

To incorporate these variables into our analysis, alongside DInSAR derived parameters, we utilized a multiclass GAM classifier. The model addressed gaps in the morphological and DInSAR based techniques, enabling the classification of a

greater number of landforms that were previously undefined in one or both inventories made by PAB and Bertone. Figure 11a visually depicts these changes, illustrating the number of features that changed classes (arrows) and those that remained within the same category (vertices). A ternary graph (Fig.11b) represents the associated probability of the rock glaciers to fall in each class. As evident from the graph, the direct class-shift from A to R (and vice versa) is a rarely frequent process (only 7 cases), and an intermediate transition passage into T class is more frequent and evident, as highlighted by the curve

trend. The observed reclassification shows that there is a common trend that transforms A into R, shifting previously through a T phase, highlighting the dynamic response of rock glaciers to environmental (fluctuations in air temperature and changes in precipitation) and geomorphological (slope orientation, ice content, debris cover) factors as described in Barsch, 1993. The transitional phase serves as a critical buffer, enabling gradual adjustments to these changing environmental drivers and facilitating smoother transitions to the relict state. This dynamic interplay is further underscored by the complex interactions

between ice presence, debris material, permafrost content, and other external factors like temperature and precipitation, often leading to non-linear responses but rather to a more transitional process (Etzelmüller et al., 2011). In our case, the transition of rock glaciers from A to R classes is also supported by velocity changes, with a decreasing trend in detected velocities processing from A to R states (Fig.11b).

This phenomenon is particularly pronounced when considering the velocity delta between the rock glaciers and their

surrounding areas not involved in the creeping process. As rock glaciers evolve towards an active state, the differential velocity between the rock glacier and its surroundings increases, indicating heightened activity and movement within the landform. This observation underscores the dynamic nature of rock glaciers and highlights the significance of velocity changes in tracking their evolution and behaviour over time.



Considering the integration of DInSAR and environmental features, a specific class may undergo reclassification when
alternative or integrated approaches are utilized. This variability underscores the importance of considering multiple factors
and methodologies in landform classification, especially in situations where input variables are incomplete or uncertain.
Moreover, relying solely on a single classification approach may be misleading, as factors such as inaccurate morphological
mapping or the inability to recognize subtle features can compromise the accuracy of the classification.

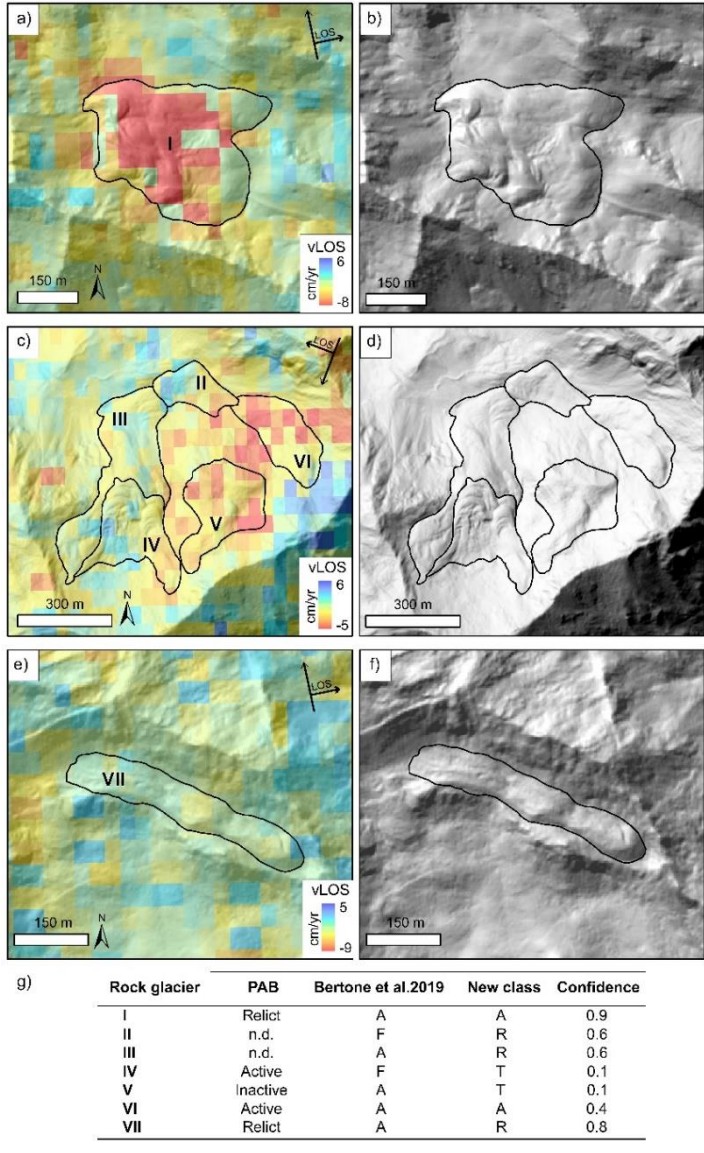

**Figure 12: Examples of rock glaciers with different activity labels. Panels a,c and d report DInSAR velocity patterns
over the selected features; b,d and f show the surface morphology through hillshade maps; g) summary table with the
original activity label for each dataset, the new class and the confidence level.**





Figure 12 shows some examples regarding the different classification among the three approaches (PAB, Bertone, and our
new approach). Rock glacier in Fig.12a,b, even if it shows the presence of swollen furrows and lobes, was classified as R by
the PAB. In Bertone and in our classification, this rock glacier is identified as A because there is both a clear velocity InSAR
signal and the presence of distinct morphological features indicate surface displacement. Opposite situation happens
considering the rock glacier IV (Fig.12c,d). This rock glacier is A for the PAB, while Bertone et al., 2019 classify it as F and
from our classification it results T. Even if this rock glacier has typical superficial structures dictated by downslope
displacements, the results derived from the coherence approach (made by Bertone) and the velocity data (from our approach)
indicate that these morphological features are presumably ''paleo'' structures, i.e. nowadays stable and therefore evidence of
a direction flow happened in the past. Relying exclusively on a single remote sensing approach that is based solely on
coherence may prove inadequate for detecting the slow movements of rock glaciers. This is especially true when these
movements do not cause noticeable changes in surface characteristics over the specified temporal baseline. Furthermore,
movement may occur primarily due to vertical deformation caused by ice melting over gentle slopes, where shear movement
does not occur. In such instances, the absence of discernible flow structures can offer valuable indications for accurately
characterizing the activity state of the rock glacier (Fig.12e.f, rock glacier VII). Other factors, such as thermal variations or
vegetation cover, may also influence activity patterns, highlighting the need for a comprehensive and diverse classification
approach to ensure accurate representation of landscape dynamics.


## 5.2 DInSAR limitations

Our results suggest that DInSAR proxies, especially the coherence statistics (Fig.5 and Fig.7), as also demonstrated by
Bertone et al., 2019, effectively discriminate the active class from the relict and inactive ones. Low coherence indicates a
diminished similarity between SAR images within the interferometric pair, typically resulting from variations in surface
scattering properties, wherein displacement emerges as a primary contributor. Conversely, high coherence values reflect
stability in target properties, signifying minimal disturbances affecting the surface of the landform. This stability results in
reduced deformation and displacement rates.

Velocity from DInSAR analysis still displays a discriminative effect, even if less sharp than coherence. This can be
attributed to the steps of the processing and filtering techniques used at a regional level, which introduce more disturbances
and might make the final velocity estimation less accurate compared to coherence. Following the specifics proposed by the
IPA group (RGIK 2023), the identification of moving areas is in fact based on the manual delineation and classification
polygons, manually drawn around InSAR pattern, usually in wrapped interferograms to have a better visualization of fringe
pattern (Bertone et al., 2022, RGIK 2023).

Given our objective to classify all mapped landforms without delving into the internal activity of individual lobes or sectors
at this stage, we opted to treat entire rock glacier polygons as moving areas and subsequently analyse their internal velocity
patterns. To speed up these analyses and facilitate application at the regional scale, we employed interferometric synthetic
aperture radar analysis utilizing Sentinel-1 data over the entire AOI. The analysis leveraged the GAMMA procedure



implemented within the HyP3 plugin on OpenSARlab, a service developed by ASF in conjunction with the MintPy package (Yunjun et al., 2019). To enhance result reliability, we iteratively repeated the time series inversion on smaller subsets of the

interferometric stack. This iterative approach facilitated the selection of reference points in closer proximity to the landforms within the AOI, ensuring thorough consideration of topographic and atmospheric conditions specific to the selected area. Despite the efficiency demonstrated by such a large-scale classification and velocity analysis approach, it is essential to acknowledge the inherent limitations associated with InSAR measurements. A significant source of uncertainty in extracting LOS velocities arises from the distance between the reference point used in the inversion and the actual landforms.

Topographic variations inherently influence error propagation, especially impacting velocity measurements as the distance from the reference point increases, particularly in regions characterized by significant elevation relief. In addition, while the mean annual vLOS provides a valuable first-order approximation, we also have to remind that it does not fully capture the 3D movement across all areas of the landform, particularly in features where multiple lobes overlap and the direction of movement diverges from the satellite vLOS. Despite this limitation, we retain the mean annual vLOS as a reference measure

within this study, recognizing its utility for large-scale classification and initial assessments of rock glacier kinematics. Previous studies (e.g., Brencher et al., 2021) have applied various methods such as reprojecting LOS measurements along the maximum slope direction or integrating both ascending and descending geometries to extract vertical and east-west movement components. However, in our approach, we chose to utilize the vLOS while taking into account the reliability index provided by the C factor (Notti et al., 2014) associated with each rock glacier. This decision was made to mitigate the

introduction of additional biases and assumptions that may arise from geometrical reprojections, while always considering the C factor to get valuable insights into the satellite's favourable orientation relative to the landform.

Another potential factor that may adversely affect the measured vLOS displacement is attributable to the CNN-APS method (Brencher et al., 2023). Since CNN methods operate directly on the data, they have the capability to filter out real portions of displacement signals rather than simply blurring them, resulting in a reduction of the displacement associated with each

feature. Consequently, while the considered vLOS provides descriptive information regarding the dynamics of each feature, this filtering effect may need to be considered when compared to the actual displacement rate.

Active phenomena in fact show displacement ranges in the order of cm/yr, while knowledge of some case studies from previous works suggests higher displacement rates exceeding tens of cm/yr. However, these are detailed specific site studies (Kofler et al., 2021, Bertone et al., 2023) where more refined DInSAR approaches, with higher resolution and control on the

area investigated, have been applied overcoming the inevitable biases associated with a regional scale problem.

Despite the general underestimation of the measured signal, related to intrinsic limitations of the SBAS approach (Pepin and Zebker, 2021) and post processing steps, the distinction between A and R features according to our model results effective as proved by the high AUC and its application to predict activity class for not defined features provided good results (Fig.8).






### 5.3 Geomorphological factors and related rock glaciers spatial distribution

After completing the classification process, a final evaluation of the classification plausibility was conducted, integrating elevation and permafrost indicators, which were initially excluded as predictor variables. This supplementary analysis confirmed that the identified patterns align with established knowledge in periglacial environments. A and T rock glaciers are typically situated at higher elevations (generally above 2600 m a.s.l.), while R classes are more commonly found at lower elevations (between approximately 2200-2500 m a.s.l.), which are consistent with widespread observations in periglacial landscapes (Fig.13a). Additionally, an assessment of permafrost occurrence probabilities (Fig.13b) within the reclassified features unveiled a significant correspondence between higher probabilities and activity classes. In fact, R is characterized by lowest elevation and lower permafrost probability respect the T and A classes, with highest elevation and more probability of preserve permafrost presence nowadays. This underscores the influential role of permafrost dynamics in shaping rock glacier activity patterns. Furthermore, the impact of lithology on controlling rock glaciers' activity is often minimal or negligible compared to the previous two factors (Fig.13c). Studies have demonstrated that lithology alone does not exert significant control over rock glaciers' activity (Kääb et al., 2005).

Given that rock glaciers primarily consist of unconsolidated debris, their movement is predominantly driven by internal deformation processes rather than lithological properties (Haeberli et al., 2006). Additionally, the insulating effect of debris cover can mitigate thermal variations in the substrate, diminishing the influence of lithological disparities on permafrost

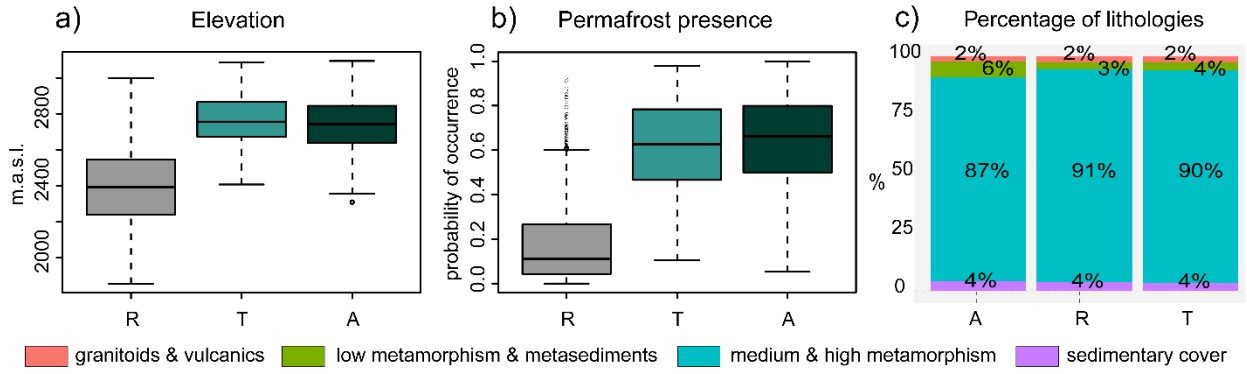

**Figure 13: distribution of elevation (a), permafrost presence (b) and percentage of lithologies (c) in the three activity classes: active (A), transitional (T) and relict (R). The lithology names indicated on the line at the bottom of the figure are referred only to the (c) panel.**

conditions and rock glacier dynamics. Therefore, while lithology may play a secondary role in modulating rock glaciers activity in specific contexts, its impact is generally overshadowed by other factors such as topography, climate, and permafrost distribution. Moreover, the need to aggregate lithologies into macro classes for regional-scale studies limits the detailed examination of their effects on rock glacier activity, highlighting the challenge of incorporating fine-scale geological variability into broader analyses.



## 5 Conclusions

This study introduces an updated classification for the state of activity of the rock glaciers in South Tyrol (Italy). The main
strength of our comprehensive approach lies in the use of replicable routines (i.e., HyP3-MintPy tools) and multivariate
statistical methods. This workflow can be adjusted and modify (for example, by selecting known stable reference points if
possible, considering different snow free months, and adopting a different atmospheric correction), and successively applied
to other areas, allowing to partially fill the gaps of the traditional techniques, morphological and dynamic classifications.
Through the integration of regional-scale spaceborne DInSAR processing with both geomorphological and climatic
descriptors, we have unified the two primary classification methods of activity of periglacial features, gathering a higher
classification spatial coverage for the mapped rock glaciers and a more robust distinction between active, inactive, and
transitional features. The integration of the kinematic information with environmental variables was accomplished through a
multiclass GAM model. This model effectively leveraged both linear and nonlinear relationships between features, providing
a statistical definition of the key variables influencing the activity classification of rock glaciers at the regional scale.
The achieved results underscore a predominance of relict features (1345 landforms mapped in total), in contrast to a
significantly smaller number of active ones (only 171). Looking at the distribution of these three classes (A, T, and R), it was
found that a transition state from active to relict rock glaciers is not a direct process. Instead, an intermediate transition phase
between A and R landforms seems to represent a common feature. At a regional scale, this transition is likely controlled by
local factors that influence not only the activity state and the evolution of rock glaciers but also affect the velocity phase of
this transition process, allowing changes from one more active class to the relict one. These local settings, characterized by
the dynamic and complex interplay of factors such as slope, lithology, and climate, shape the dynamics of rock glaciers,
leading to varying rates of progression between different states of activity.

## Competing interests

The contact author has declared that none of the authors has any competing interests.

## Acknowledgements

This study was funded by the European Union - NextGenerationEU, in the framework of the consortium iNEST -
Interconnected Nord-Est Innovation Ecosystem (PNRR, Missione 4 Componente 2, Investimento 1.5 D.D. 1058 23/06/2022,
ECS_00000043 – Spoke1, RTx, CUP I43C22000250006). The views and opinions expressed are solely those of the authors
and do not necessarily reflect those of the European Union, nor can the European Union be held responsible for them.

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
