# Peer review of "Optimizing rock glaciers activity classification in South Tyrol (North-East Italy): integrating multisource data with statistical modelling"

_EGUsphere, 2024_

## Referee Comment (RC2)

Title: Optimizing rock glaciers activity classification in South Tyrol (North-East Italy): integrating multisource data with statistical modelling

**1) Overall quality and general comments**

Rock glaciers are key indicators of permafrost in alpine regions, formed by a seasonally frozen detrital layer overlying supersaturated debris of ice or pure ice, and characterized by gravity flow. Their distribution is influenced by topographic and climatic factors at different scales, and they play a crucial role in high-altitude hydrology by storing ice and water. Traditionally, rock glaciers are classified as active, inactive, or relict based on ice content and movement. However, rising permafrost temperatures have led to an accelerating trend, encouraging an updated classification that considers sediment transport efficiency. In the regional territory of South Tyrol, two rock glaciers activity classifications coexist (Autonomous Province of Bolzano/Bozen and Bertone et al., 2019). By combining geomorphological characteristics, climatic driving factors, and InSAR products, the authors develop a statistical model to refine the classification of rock glaciers.

This study represents an innovative contribution since it integrates multiple variables into a multiclass generalised additive mixing (GAM) model to predict rock glacier activity. Using remote sensing, ground-based data, and digital terrain models, the workflow involves extracting velocity and environmental attributes at a regional scale, calibrating and validating a multiclass predictive GAM, and applying it to classify landforms based on their activity status.

The integration of remote sensing data and statistical modelling significantly advances current methods for assessing rock glacier dynamics. The study is well-structured, with a clear research objective and methodology. The statistical approach, particularly the use of a multiclass GAM model, is effective for the research aims. The discussion is robust, highlighting both its contributions and its limitations. The figures and tables are clear, informative, and support the understanding of the concepts. Finally, this work advances the understanding of rock glacier dynamics by refining their classification system and linking their activity states to a range of predictor variables.

**2) Individual scientific questions**

3.4.1. Statistical modelling

How did the authors ensure the robustness of the GAM model in terms of the selection and evaluation of predictor variables?

4.1. Exploratory Data Analysis

How did you decide which variables to retain for further analysis, and why were some variables, such as elevation, excluded to avoid redundancy despite their high discriminatory power? Could you clarify the rationale behind this choice?

**3) Specific comments on the manuscript**

*Line 119:* How many rock glaciers are present in the analysed dataset?

*Line 127:* The classification 'n.d.' is unclear. Could you please clarify its meaning and usage in this context?

*Line 148:* Could you explain in more detail how the variables were extracted and assigned to each individual rock glaciers?

*Lines 207-210:* "Using this SCD parameter, a potential correlation between the rock glaciers' activity at a regional level was made[...]" Could you explain this statement more clearly? How was the correlation assessed, and what were the main findings regarding the SCD in relation to the rock glaciers' activity?

*Figure 4*: Does the term "look vector" refer to the Line of Sight (LOS) of the satellites? Could you also better explain if the shadowing and layover effects part is the C index analysis?

*Figure 4:* Is the vLOS referring to vertical velocity? Additionally, could you adjust the colour scale bar to range between -8 and 8 cm/year to improve the clarity of the data representation?

*Lines 244-248:* "For each rock glacier polygon, mean values for environmental and climatic variables were assigned based on the values within the polygon boundary. Furthermore, for DInSAR-related variables (i.e., velocity and coherence), additional statistical descriptors [...]". Can you explain how the uncertainty was computed for each rock glacier, based on the SAR data coverage? How did you assess the spatial uncertainty within each polygon?

*Line 243*: "Starting from the distribution map of the rock glaciers and considering their displacement range, we made two distinctions [...]". Could you clarify the rationale behind the choice of a 100-meter buffer around each mapped landform? How was this distance determined, and how does it affect the classification?

*Lines 264-266*: "To discern the key factors influencing the distinction between A, R, and T rock glacier classes, we performed an initial Exploratory Data Analysis. This exploration served [...]". Could you provide more details on how this exploratory analysis was performed, and how it helped with the model?

*Lines 167-272:* "GAM was employed to investigate the associations between the chosen predictor variables derived from both environmental and DInSAR datasets and the response variables. GAM provides [...]". Could you provide more explanation on the use of GAM in this context? A brief discussion of the relevant literature and how GAM has been applied in other studies would strengthen this section.

---

## Author Comment (AC1)

**Responses to the comments of anonymous reviewer #1**

In this paper, the authors integrate existing rock glacier results from South Tyrol region, including the Autonomous Province of Bolzano/Bozen (PAB) rock glacier inventory data and the DInSAR-derived movement status by Bertone et al., (2019). By combining geomorphological characteristics of the rock glaciers, climatic driving factors, and InSAR products, the statistical model is calibrated and validated. This model is then used to optimize the identification of A (Active), T (Transitional), and R (Relict) states of rock glaciers in the region, while also describing the relationship between rock glacier states and multiple driving variables.

**General comments**

1. The reliability of optimizing rock glacier state. The author combines the PAB (2010) and Bertone (2019) rock glacier inventories in South Tyrol and uses various statistical factors to setup the GAM model, which is then applied to optimize the rock glacier states in the study area. However, it should be noted that if the original rock glacier states are not entirely accurate, the resulting GAM will inevitably carry uncertainties, making it inappropriate to use such a model to further optimize these states. Further assessment of the classification accuracy should be implemented or discussed.

We fully agree that starting from an initial classification with inaccuracies can introduce uncertainties into the results derived from the GAM (Generalized Additive Model). Recognizing this, we have taken a cautious approach to minimize these uncertainties.
In our initial reclassification (Table 1), we adopted a conservative strategy to establish a reliable baseline:
- Landforms classified as "active" were included only if they were consistently categorized as active by both methods.
- Similarly, landforms showing no movement were classified as "relict."

This selective approach ensures that the initial dataset used to train the GAM is as robust and accurate as possible. The GAM is then employed to refine and improve this initial classification by incorporating additional morphometric parameters and velocity descriptors. By integrating these complementary datasets, we aim to enhance the reliability and comprehensiveness of the classification process.
Discussions on this aspect are provided in the results and validation sections to highlight the model's reliability and areas for potential refinement.

2. In Section 3.4, it is unclear on the training data used to support the setup of GAM model. How many rock glacier samples were used and treated to train the model, including the samples for each status type (A, T, R), should be clearly presented.

In our analyses we applied the GAM model on the entire dataset of transitional, relict and active features (after reclassification in table 1) without splitting it for training and testing. This can be justified by the nature of the dataset and the objectives of the study. The initial dataset was already subject to a preliminary reclassification (Table 1), which we recognized as an approximation rather than a definitive classification. Given this context, our goal was to refine and improve upon this initial classification by leveraging the relationships between the parameters identified through the GAM.

Using the full dataset allowed us to maximize the information available for extracting these parameters relationships and constructing a predictive model capable of reclassifying all features into more reliable and meaningful activity classes. A predicted probability was also assigned to each landform, providing a quantitative measure of the confidence and reliability of the classification.

To ensure the robustness and generalizability of the results, we also evaluated the predictive model using k-fold cross-validation with varying sample sizes (2-fold, 5-fold, and 10-fold). This step allowed us to test the model's performance across different subsets of the data, ensuring that the reclassification was not overfit to the full dataset. The use of multiple cross-validation strategies further validates the consistency and reliability of the model, demonstrating that the relationships and classifications derived from the full dataset are representative and robust.

3.  The paper uses SBAS-InSAR products, including velocity and coherence, as inputs to implement the statistical model, which is a distinctive aspect of the study. However, when the movement rate map of existing rock glaciers is available, classification can be conducted directly from a kinematic perspective (RGIK, 2023): for example, 10 cm/yr > v > 1 cm/yr for T (Transitional), v > 10 cm/yr for A (Active), and v < 1 cm/yr for R (Relict). This quantitative description of rock glacier movement status would be more straightforward. Moreover, if the prerequisite for identifying rock glacier states is to perform DInSAR or SBAS-InSAR to obtain product input for the statistical model, it further limits the application of rock glacier analysis in mountainous regions. To comprehensively define rock glacier states by integrating climatic driving factors, displacement rates, and morphological parameters, it is advisable to compare the results obtained from the current comprehensive definition with the states identified solely based on movement velocity data, or to use empirical data to demonstrate that the states identified by this comprehensive method are more accurate.

We appreciate the reviewer's comment regarding the use of velocity maps for directly classifying rock glaciers activity. We acknowledge that velocity-based classification offers a straightforward and quantitative approach. However, as the reviewer rightly noted, obtaining reliable and comprehensive velocity maps over mountainous regions is challenging. Issues such as topographic masking and low coherence often hinder the generation of accurate velocity data using DInSAR or SBAS-InSAR techniques.

To address these limitations, we have integrated morphometric and velocity attributes into our methodology. By combining these two approaches, we aim to overcome the weaknesses inherent in relying solely on velocity data. This integration allows for a more robust and comprehensive classification of rock glaciers activity.

Our approach has demonstrated clear advantages. Specifically:
- Expanded Classification: We successfully classified more polygons than the existing dataset, leaving only 63 out of 1779 landforms unclassified. Previously, considering only the morphological classification 235 rock glaciers were not classified, while 331 remained unclassified considering the coherence-based approach.
- Alignment with Existing Classifications: The results show that 67% of activity labels remain unchanged compared to previous classifications. The 32% of the old polygons were reclassified into different categories.

In conclusion, although velocity maps offer a direct means of classifying rock glaciers, our integrated approach overcomes the practical challenges of obtaining accurate velocity data in wide and difficult terrains. Furthermore, it improves classification accuracy by incorporating complementary datasets, resulting in a more thorough and consistent understanding of rock glacier dynamics.

**Specific comments**

1. Line 28: "Our approach improved classification accuracy, leaving only 3.5% of features unclassified compared to 13% in morphological classification and 18.5% in DInSAR-based methods." If "feature" here refers to the number of rock glaciers, such as the 3.5% representing 63 out of a total of 1,779, then it doesn't represent accuracy. Instead, it should be considered an enhancement over previous work, providing a more complete or comprehensive cataloging of rock glacier states.

We agree with the reviewer on this observation. We modified the sentence accordingly.

2. Line 60: "Although widely used, this classification brings two relevant limitations both from subjectivity point of view (activity attribution based on geomorphological approach is depended on the operator skills) as well as categorization since the activity of rock glaciers is considered constant over time at the scale of decades to centuries." The classification into "intact" and "relict" does not inherently introduce subjectivity; rather, it is the geomorphological classification process that carries subjective factors. If we classify Active (A), Relict (R), and Transitional (T) states based on geomorphological characteristics, it would also involve subjectivity. The use of initial identification data, such as the x-axis in Table 1, which is also based on geomorphological characteristics, introduces subjectivity into the GAM as well. Please explain why the assumption of long-term invariability in activity state identification would be considered a limitation of the "relict" and "intact" classification.

The sentence at line 60 was ambiguous and therefore was changed. However, in our view, classifying rock glaciers as intact or relict is based on the presence of inner permafrost, and without subsurface data, this classification remains somewhat subjective. Of course, this is also true for the classification in active, relict and transitional, but this latter classification is not merely based on the presence or lack of permafrost, instead it takes into account several potential descriptive factors that, if used together, could provide a more comprehensive description of the activity state of each landform.

3. Line 125: Bertone et al., (2019)

Added

4. Lines 193-199: What role does precipitation play in the overall text? Precipitation was not included as an input in the GAM; is it meant to be part of the discussion on precipitation? However, there doesn't appear to be any statistical information provided to support the author's discussion on precipitation.

We thank the reviewer for this observation and agree that precipitation is an important factor influencing rock glacier activity. As noted in the discussion section (lines 464-471), precipitation was not included as an input in the GAM due to the absence of a clear correlation between mean summer and winter precipitation values and activity classes at the regional scale.

While precipitation events are known to contribute to short-term and seasonal variations in rock glacier velocity (Kenner and Magnusson, 2017; Kenner et al., 2021), our analysis highlights that such relationships are less evident at broader regional scales. This is likely due to the influence of local factors, such as temperature, aspect, and other site-specific conditions, which can modulate the impact of precipitation on rock glacier dynamics.

Furthermore, as discussed in Zhang et al. (2023), precipitation interacts strongly with other climatic and environmental factors, making it challenging to isolate its effect on activity classes. For these reasons, further investigation is required to quantitatively understand the relationship between precipitation and rock glaciers activity at regional scales. Thus, precipitation was excluded from the current analysis but remains an important avenue for future studies.

5. Section 3.3: How were the velocity datasets from ascending and descending InSAR results integrated? Why was the calculation of slope velocities for rock glaciers not performed, given that both ascending and descending InSAR maps were derived?

We thank the reviewer for his observation regarding the integration of ascending and descending InSAR datasets and the decision not to calculate slope velocities for the rock glaciers. The ascending and descending datasets were analyzed by selecting independently the most suitable acquisition for each landform according to its topographic orientation.

The decision to not calculate slope-parallel velocities ($V_{SLOPE}$) was made in order to avoid reprojection and consequently reduce the uncertainties associated with this process. Specifically, reprojection requires dividing the line-of-sight (LOS) velocities by the C factor and, as noted by Notti et al. (2014), this process introduces significant uncertainty, particularly for values of $-0.2 < C < 0.2$, where reprojection becomes unreliable. Additionally, reprojection assumes that all motion occurs as pure downslope sliding, which oversimplifies the complexity of rock glacier dynamics. These considerations, as stated in lines 547–550, guided our decision to rely on LOS velocities instead of slope-parallel velocities for the analysis.

6. Line 253: Is the 100m value an empirical choice? If a unit within a rock glacier system is entirely occupied by other units within a 100 m buffer zone, how is this situation handled? I agree with the author's idea of calculating the increment by comparing the values inside and outside the buffer zone. This increment can potentially distinguish between the rock glacier's intrinsic movement and movement caused by external factors. It seems that further analysis or application of this increment has not been addressed in the following sections.

The selection of a 100 m buffer was chosen since it provides a balance that ensures meaningful data extraction for analysis while avoiding excessive noise from unrelated features.

In cases where adjacent or coalescing rock glaciers occur, the rims were cut to avoid any overlap between features, ensuring that the boundaries of one rock glacier do not encroach upon another. Additionally, due to differences in the orientation and spatial distribution of the landforms, it is highly unlikely for an entire rock glacier to fall entirely within the 100 m buffer zone of another.

Regarding the suggestion for further analysis of the increment values: we concur that differentiating between intrinsic movement and externally driven movement is an interesting prospect. However, this aspect was not explored further in the current study, as our primary focus was on classifying rock glaciers activity. The incremental differences derived from buffer-based analysis could indeed serves as a basis for future investigations into the dynamics of rock glacier systems, particularly in distinguishing between intrinsic and external movement drivers.

7.  Line 285: Cross-validation is generally used because the data is limited, and it helps improve the model's generalization capability. It also allows for better evaluation and enhances the model's ability to fit data outside the training set. Please clarify the rationale for consecutively using 2-fold, 5-fold, and 10-fold cross-validation.

In this study, we applied 2-fold, 5-fold, and 10-fold cross-validation consecutively to provide a more robust and comprehensive validation of our model. The rationale behind this approach was to ensure that the model's performance metrics remained consistent across different levels of data partitioning regardless of the chosen partitioning method.

8.  Line 309: From the boxplot (Figure 5c), it appears that VRM (Vector Ruggedness Measure) doesn't show a significant signal, which might suggest that surface roughness is unrelated to the activity status. Therefore, the inclusion of VRM in the GAM model seems unjustified. There are many other potential factors that could serve as surface condition indicators, such as terrain curvature and vegetation cover.

We agree with the reviewer on this point. Curvature was not included in the analysis because our exploratory data analysis (EDA) did not reveal clear differences between the classes based on this parameter (boxplots below). Similarly, as noted by the reviewer, the VRM does not show a sharp distinction between classes, although the median value is slightly higher for the transitional and active classes compared to the relict class (Figure 5c). Despite this, we chose to retain VRM as a parameter because it is more closely related to local-scale surface irregularities, which, in our opinion, makes it a relevant descriptor of activity. While the differences are not particularly pronounced due to the scale and heterogeneity of our mapped landforms, we believe that VRM still offers useful information for understanding the activity levels of rock glaciers.

[Figure]

9.  Section 4.2: Were the normalized or raw values of the eight variables used in the GAM model? How many rock glacier samples were used? Is it the number of A+T+R as mentioned in Table 1? Additionally, how did the author deal with the rock glaciers located in the layover/shadow regions of the SAR data?

We applied the GAM model to the entire dataset, which included all features with a preliminary classification attribute (1334 features, as detailed in Table 1) categorized as relict, active, or transitional. This approach aligns with the reasoning provided in response to General Comment 2, where we justified using the full dataset to refine the initial approximate classification.
For rock glaciers located in topographically unfavorable areas affected by layover and shadowing in the SAR data, we addressed these challenges by excluding only the pixels impacted by these effects. The features themselves were not excluded from the analysis; instead, their DInSAR data points were

reduced in number. This approach allowed us to retain as much useful information as possible while mitigating the influence of these data gaps on the overall analysis.

10. Line 383: The GAM is trained as a classifier based on the environmental factors of the original rock glacier data and the DInSAR products. However, it seems unreasonable to apply the trained GAM model to all 1,779 rock glaciers in the region, including those initially used as training datasets.

To clarify this point, we refer to our reply to the second general comment.

11. Lines 465-472: The paper lacks information on the statistical relationship between precipitation and rock glacier activity status.

We performed an exploratory data analysis on the relation between precipitation and rock glaciers activity, but we found no interesting correlation. We thus not included this variable in the GAM. We discussed our decision in lines 464-471.

[Figure]

12. Lines 500-507: The current method for evaluating identification accuracy involves both InSAR movement signals and distinct morphological characteristics, providing a quantitative perspective first and then a subjective morphological identification perspective. However, if the current GAM identification method requires InSAR products, such as movement velocity and coherence as inputs, why not directly use the velocity map along with geomorphological features to evaluate the status?

Thank you for the comment. By integrating both velocity data and morphometric features into the GAM, we are indeed performing the type of analysis the reviewer is suggesting. The GAM approach inherently combines quantitative InSAR-derived velocity metrics and coherence with morphological characteristics to evaluate and classify rock glacier states.
This method is designed to leverage the strengths of both datasets: velocity maps provide direct kinematic information, while morphological parameters offer additional insights that may compensate for gaps or inconsistencies in InSAR data caused by factors such as topographic masking or low coherence. Thus, our approach moves beyond relying solely on a velocity map by

incorporating complementary geomorphological data, ensuring a more robust and comprehensive classification.

13. Lines 541-551: $V_{los}$ cannot fully represent the true movement pattern of rock glaciers. Could this limitation affect the uncertainty of the GAM? There might be cases where some rock glaciers have a large $V_{los}$ but a much smaller actual movement rate, thus introducing uncertainty.

All the uncertainties and errors associated to the measure of a certain variable indeed propagates in the GAM model. However, we can only limit them and not totally remove them. So, it is true that the underestimation of the movement can affect the GAM, but this is an intrinsic limitation we know we have to take into account. For this reason, we filtered out noisy component from the velocity data. We just point out two examples for a clear explanation: (i) removal of shadow and layover areas or/and (ii) consider for each rock glacier the acquisition geometry that better catch movements over that slope.

Furthermore, we have few or none examples of rock glaciers exhibiting a large vLOS coupled with a small actual movement rate. In contrast, it is more common to observe the opposite: large displacement rates that are underestimated when measured along the LOS.

**Comments on Figures**

- Figure 2: The "calibration" step in the "multiclass GAM model" is vague. Typically, calibration involves adjusting something that was previously incorrect to make it correct. Could you clarify what this step entails in the context of your model?

In this context, model calibration refers to identifying a set of parameters that accurately describe the system's behavior. Specifically, it involves finding parameters that are relevant for describing the activity of rock glaciers.

- Figure 3: The units of the unwrapped phase should be "rad" rather than "cm/yr."

The unwrapped phase has been here already converted in cm/yr in the unwrapping procedure.

- Figure 4: Do the input dataset "look vectors" correspond to Figure 4b (the visibility map)?

The look vectors refer to the parameters of LOS and azimuth angles components that are considered to produce the map in fig 4b.

- Figure 5: How were the outliers identified, and how was the lower limit chosen, especially for the coherence, such as in Figure 5d? The median and quartile changes with R, T, A are quite reasonable, but I've noticed that there are many outliers close to your lower limit. Please explain how the lower limit was determined and why there are so many outliers, not just in Figure 5d. Also, why not calculate and present the "mean velocity," "variance of velocity," and "velocity outside delta (Δ)" plots like the coherence panel?

In the boxplot function the whiskers and box dimensions are defined: the lower edge corresponds to the 25th percentile (Q1), the middle line is the median, the 50th percentile, and the upper edge the 75th percentile (Q3). The whiskers extend to the most extreme data points that are not considered

outliers. This range is considered between 1.5*IQR from Q3 and 1.5*IQR from Q1. Beyond these limits points are considered outliers.

The high number of outliers in the coherence-related plot, particularly in Figure 5d, is largely due to the inherent characteristics of SAR imagery. These data are influenced by various independent decorrelation factors such as imaging geometry, processing artifacts, and thermal noise. On rock glacier surfaces, which are composed of heterogeneous materials and characterized by varying movement velocities, coherence values can be highly scattered. This variability results in both wide variance and a significant number of outliers. Particularly, relict features often exhibit slightly rolling blocks due to scree movement or surface disturbances, which sporadically decrease in coherence and contribute to these outliers.

Regarding the exclusion of mean velocity and velocity variance in the plot, we opted not to include mean velocity because it does not adequately represent displacement rates. Averaging positive and negative velocities could significantly underestimate the magnitude of movement for various features. Instead, we chose to focus on the highest negative velocity, as it provides a more accurate representation of displacement rates without averaging opposing values. The delta of velocity, while not included in this plot, was considered in Figure 11, where we demonstrated how the delta increases between active and relict features.

- Figure 5h: Is a velocity threshold (0.02 m/yr) being applied?

No, no filtering on velocity is here applied.

- Figure 6f: Although the relationship may not be immediately apparent, I have observed that many rock glaciers are frequently located in convergent areas. This pattern is intriguing, and any insights or explanation regarding this observation would be valuable.

We think this is mainly due to the fact that rock glaciers are located in valleys, so convergent areas where debris from rockwalls are channeled and accumulated.

- Figure 11b: Would replacing the LOS velocity values with slope velocity result in a smoother transition from red to blue?

In Figure 11b, we do not use LOS velocity values directly; instead, we present the velocity delta, which is calculated as the difference between the velocity of the rock glacier and its surrounding rim. This approach was chosen because the delta better captures relative movement dynamics, highlighting the distinction between the rock glacier and its immediate surroundings.

In our analysis, we did not compute velocity along the slope for reasons outlined in response to comment 5. Using the 1D LOS velocity alone did not sufficiently explain the observed differences in activity levels or provide the same clarity as the delta approach. The velocity delta offers a more meaningful metric for distinguishing activity states, as it accounts for both the internal dynamics of the rock glacier and its interaction with the surrounding environment. As a result, we believe this method provides a more nuanced depiction of the transition between active and less active states.

---

## Author Comment (AC2)

**Responses to the comments of anonymous reviewer #2**

**1) Overall quality and general comments**

Rock glaciers are key indicators of permafrost in alpine regions, formed by a seasonally frozen detrital layer overlying supersaturated debris of ice or pure ice, and characterized by gravity flow. Their distribution is influenced by topographic and climatic factors at different scales, and they play a crucial role in high-altitude hydrology by storing ice and water. Traditionally, rock glaciers are classified as active, inactive, or relict based on ice content and movement. However, rising permafrost temperatures have led to an accelerating trend, encouraging an updated classification that considers sediment transport efficiency. In the regional territory of South Tyrol, two rock glaciers activity classifications coexist (Autonomous Province of Bolzano/Bozen and Bertone et al., 2019). By combining geomorphological characteristics, climatic driving factors, and InSAR products, the authors develop a statistical model to refine the classification of rock glaciers.

This study represents an innovative contribution since it integrates multiple variables into a multiclass generalised additive mixing (GAM) model to predict rock glacier activity. Using remote sensing, ground-based data, and digital terrain models, the workflow involves extracting velocity and environmental attributes at a regional scale, calibrating and validating a multiclass predictive GAM, and applying it to classify landforms based on their activity status.

The integration of remote sensing data and statistical modelling significantly advances current methods for assessing rock glacier dynamics. The study is well-structured, with a clear research objective and methodology. The statistical approach, particularly the use of a multiclass GAM model, is effective for the research aims. The discussion is robust, highlighting both its contributions and its limitations. The figures and tables are clear, informative, and support the understanding of the concepts. Finally, this work advances the understanding of rock glacier dynamics by refining their classification system and linking their activity states to a range of predictor variables.

**2) Individual scientific questions**

**1.1. ) 3.4.1. Statistical modelling**

How did the authors ensure the robustness of the GAM model in terms of the selection and evaluation of predictor variables?

The selection of predictor variables for the GAMs model was conducted through exploratory data analysis, which enabled the screening of a broad set of morphometric and climatic descriptors (Table 2). From this analysis, eight variables were chosen based on the interquartile ranges that exhibited the greatest divergence among classes.

**1.2. ) 4.1. Exploratory Data Analysis**

How did you decide which variables to retain for further analysis, and why were some variables, such as elevation, excluded to avoid redundancy despite their high discriminatory power? Could you clarify the rationale behind this choice?

From the exploratory data analysis, we selected variables that exhibited the greatest interquartile variations in boxplot distributions, as these differences enhance class discrimination. Some variables, such as elevation, were excluded despite their discriminatory power because they are strongly correlated with retained predictors, like land surface temperature (LST). Similarly, aspect and total insolation influence LST and were excluded to avoid redundancy, as their contribution is already captured through LST. This helps minimize redundancy of information, ensuring a more efficient and interpretable mode.

**3) Specific comments on the manuscript**

*2.1. ) Line 119: How many rock glaciers are present in the analyzed dataset?*

The dataset used includes 1779 features. This information is reported at line 127 in the manuscript. We modified the sentence to better clarify this point.

*2.2. ) Line 127: The classification 'n.d.' is unclear. Could you please clarify its meaning and usage in this context?*

We added the definition of "n.d." that stands for "not defined".

*2.3. ) Line 148: Could you explain in more detail how the variables were extracted and assigned to each individual rock glaciers?*

Morphometric and terrain attribute analyses were conducted using ArcGIS 10.8 and SAGA GIS, based on a 10m DEM resolution. All derived products (e.g., slope, and aspect) were generated for the entire South Tyrol region and successively clipped over the boundary of each rock glacier presents in the dataset (1779 in total). For each feature, we calculated the mean values of environmental and climatic variables. Additionally, for the DInSAR-derived variables, we computed further statistical descriptors, including variance and the 25th, 75th, and 90th percentiles, to better capture their internal variability. The details of these analyses are provided in Section 3.4.1.

*2.4. ) Lines 207-210: "Using this SCD parameter, a potential correlation between the rock glaciers' activity at a regional level was made[...]" Could you explain this statement more clearly? How was the correlation assessed, and what were the main findings regarding the SCD in relation to the rock glaciers' activity?*

We thank the reviewer for this comment and agree on the scarce clarity of the sentence. To enhance understanding, we have revised it in the text accordingly (Lines 206-210): "we do not consider SCD as a predisposing factor for the development of rock glaciers due to its implications for the thermal state of permafrost. Instead, we consider the temporal duration of snow cover in relation to the observed activity of rock glaciers, viewing SCD primarily as a factor influencing the modulation of activity states rather than as a prerequisite for their onset."

*2.5. ) Figure 4: Does the term "look vector" refer to the Line of Sight (LOS) of the satellites? Could you also better explain if the shadowing and layover effects part is the C index analysis?*

Look vectors do not correspond directly to the LOS. They are the component of a 3D directional vector from the ground back to the sensors and they are described by two angles: the look vector elevation angle and the look vector orientation angle. The first measures the angle between the look

vector and a horizontal plane at the ground pixel and indicates the sensor position above the surface. The latter is defined as the angle between the East direction and the projection of the look vector on the horizontal surface plane. These angles are considered in combination with the DEM to highlight the areas that, due to topographic and geometric conditions, are affected by layover, shadowing and foreshortening. C index is related to the evaluation of the visibility for each landform but can be better interpreted as representative of the percentage of movement detected from the satellite on the ground. So, after the exclusion of layover and shadowing areas, we used it as parallel information to quantify the robustness of the SAR measurement over each rock glacier. Rock glaciers expose N-S have a lower C value in comparison to those which have more favorably oriented towards east and west.

> 2.6. ) *Figure 4: Is the vLOS referring to vertical velocity? Additionally, could you adjust the color scale bar to range between -8 and 8 cm/year to improve the clarity of the data representation?*

vLOS does not refer to the vertical velocity, but to the velocity component along the line of sight. We did not compute the vertical velocity, but kept the 1D LOS information because, as explained in lines 549-554, we prefer to mitigate the introduction of biases and assumptions that may arise from geometrical reprojections.

We modified the color scale.

> 2.7. ) *Lines 244-248: "For each rock glacier polygon, mean values for environmental and climatic variables were assigned based on the values within the polygon boundary. Furthermore, for DInSAR-related variables (i.e., velocity and coherence), additional statistical descriptors [...]". Can you explain how the uncertainty was computed for each rock glacier, based on the SAR data coverage? How did you assess the spatial uncertainty within each polygon?*

The spatial uncertainty within each rock glacier polygon is not quantified by a single index, but it is assessed by evaluating the SAR data coverage and quality, also adding a filter on coherence (>0.25) and velocity (±2mm/yr). The C-map is also used to indicate the satellite's detection capability for each rock glacier, highlighting areas where signal coherence and data reliability might be reduced. Furthermore, we filtered the satellite data to exclude regions affected by layover and shadowing, ensuring that only valid pixels were included in the analysis.

> 2.8. ) *Line 243: "Starting from the distribution map of the rock glaciers and considering their displacement range, we made two distinctions [...]". Could you clarify the rationale behind the choice of a 100-meter buffer around each mapped landform? How was this distance determined, and how does it affect the classification?*

The selection of a 100 m buffer was chosen since it provides a balance that ensures meaningful data extraction for analysis while avoiding excessive noise from unrelated features. In cases where adjacent or coalescing rock glaciers occur, the rims were cut to avoid any overlap between features, ensuring that the boundaries of one rock glacier do not encroach upon another. Additionally, due to differences in the orientation and spatial distribution of the landforms, it is highly unlikely for an entire rock glacier to fall entirely within the 100 m buffer zone of another. Regarding the suggestion for further analysis of the increment values: we concur that differentiating between intrinsic movement

and externally driven movement is an interesting prospect. However, this aspect was not explored further in the current study, as our primary focus was on classifying rock glaciers activity. The incremental differences derived from buffer-based analysis could indeed serves as a basis for future investigations into the dynamics of rock glacier systems, particularly in distinguishing between intrinsic and external movement drivers.

> 2.9. *) Lines 264-266: "To discern the key factors influencing the distinction between A, R, and T rock glacier classes, we performed an initial Exploratory Data Analysis. This exploration served [...]". Could you provide more details on how this exploratory analysis was performed, and how it helped with the model?*

We refer to our response to comment 2.2. The Exploratory data Analysis (EDA) was conducting analyzing the distribution of statistical descriptors of morphometric, climatic and DInSAR derived parameters in all the mapped features grouped in the three main activity classes. This step is fundamental to extract the most representative variables controlling the distinction between A, R and T rock glaciers. In GAM, using less significative parameters as predictor variable would provide less sharp classification with associate a lower prediction capability. We thus considered a group of variables, eight in total, that have a physical control on the activity (e.g. LST, SCD) or are direct consequences of it (VRM, velocity etc.) and that, at the same time, provide a statistical distinction between activity classes.

> 3.10) *Lines 267-272: "GAM was employed to investigate the associations between the chosen predictor variables derived from both environmental and DInSAR datasets and the response variables. GAM provides [...]". Could you provide more explanation on the use of GAM in this context? A brief discussion of the relevant literature and how GAM has been applied in other studies would strengthen this section.*

We thank the reviewer for this suggestion. We have integrated the text with additional references on the application of GAM in similar studies. We selected GAM over a linear model because it can effectively capture complex, non-linear relationships between response variables and multiple independent environmental predictors. This flexibility is particularly important in our study, where the relationships between geospatial and DInSAR-derived variables may not follow a simple linear trend. By employing GAM, we ensure a more accurate representation of the underlying associations in our dataset.

---

## Author Response (AR2)

**Egusphere-2024-1511**

Optimizing rock glaciers activity classification in South Tyrol (North-East Italy): integrating multisource data with statistical modelling

Chiara Crippa*, Stefan Steger, Giovanni Cuozzo, Francesca Bearzot, Volkmar Mair, and Claudia Notarnicola

**Response to the editor's comments:**

**General comments**

*1.      There is some similarity (e.g. similar study region, InSAR velocity calculation; rock glacier classification, LST calculation) with the study by Agliardi et al., (2024) where the main authors of this study is co-author of and which was submitted to TC at a similar time. The similarities and differences to this study need to be better shown. Also, the reviewers of the Agliardi et al. (2024) study were critical. Clarify the impacts to this study.*

We acknowledge the reviewer's observation regarding the similarities with the study by Agliardi et al. (2024). While both studies address the broad topic of rock glacier activity and share some methodological elements, such as the use of InSAR velocity data and LST as relevant environmental parameters, the two works are fundamentally distinct in several key aspects. First, the studies focus on different geographic areas and datasets. The current study is specifically centered on the South Tyrol region and is part of the PNRR-funded iNEST project, which has the explicit objective of investigating high mountain hazards in this area. In contrast, the Agliardi et al. study targets a different region (the uppermost portion of Valtellina, IT) and was developed independently. Second, the methodological approaches differ significantly. While Agliardi et al. rely primarily on wrapped interferograms for their InSAR analysis, our study employs unwrapped interferograms integrated in a SBAS procedure. Moreover, we enhance our approach by incorporating statistical modeling through Generalized Additive Models (GAMs), which offers a novel and more flexible interpretation of the controls on rock glacier dynamics. Although some variables such as LST are common between the two studies, this choice is based on their relevance to periglacial processes and not on methodological overlap. Thus, aside from addressing a broadly similar research theme, the two studies are distinct in their aims, datasets, methodologies, and spatial contexts.

*2.      I find the abbreviations A, T, R a bit confusing (T is usually used for temperature etc.). E.g. readers that first read the abstract and conclusions and look at the figures won't understand. These are also uncommon and not needed as there is no character limitation with TC. I suggest to write in full, but in italic.*

As suggested, we changed the abbreviations through the text with the italic extended words. We only kept some abbreviations in the figures for sake of simplicity and a better representation but integrated the captions accordingly.

3. *The snow cover duration is important as also your analysis shows. However, the temperature condition when the snow cover starts to develop is also quite important to consider as is the snow thickness (in particular for the blocky material on rock glaciers). You may want to discus this at least with some sentences.*

We agree that these parameters play a crucial role in controlling ground thermal regimes and, consequently, permafrost dynamics. However, in the present study, being on a regional scale, we did not have access to widespread in-situ ground temperature data or detailed snow thickness measurements across the study area. As such, our analysis focuses on remotely sensed variables such as snow cover duration and land surface temperature (LST), which, while informative, do not fully capture the thermal insulation effects of snowpacks or the timing and nature of early-season snow cover.

Nonetheless, land surface temperature is indirectly accounted for in the snow cover duration (SCD) retrieval approach, as detailed in Notarnicola et al. (2013). Specifically, temperature thresholds derived from MODIS band 31 (11 µm) are used to filter out snow-free areas, thereby improving the accuracy of snow cover detection. These thresholds are seasonally adjusted (e.g., 283 K for winter, 290 K for summer) to reflect variations in surface temperature and to minimize false snow detections, especially in transitional periods and over complex terrain.

The reference to this work and the explanation for why we do not consider snow thickness or temperature can be found in lines 200–205.

**Specific comments:**

4. *L12: I suggest avoiding the term "climate warming". Although frequently used it is physically not correct. You may think about writing "atmospheric warming". More important: write "clear signs" and not "the clearest". The polar cryosphere shows also clear signs of degradation.*

We agree with the Editor on these points. We updated the text as suggested.

5. *L36: I suggest referring also to another reference for rock glaciers (e.g. Berthling et al. 2011) or from encyclopaedia entries (e.g. Kääb 2013 or Janke & Bolch, 2022).*

We thank the editor for the suggested references. We integrated them in the text and in the bibliography.

6. *L78: "a" digital terrain model. And omit "to derive" as it refers only to the DTM bit not to the other data or add the info what variables are derived from the satellite data etc. for consistency.*

We accept the suggestion and modify the sentence as: "We derived the input variables by integrating multiple data sources, including multispectral satellite imagery (Landsat, MODIS), radar data (Sentinel-1), interpolated ground measurements from weather stations, and variables extracted from digital terrain model (DTM) analysis.

7.     *L95ff: Please add the info about the elevation the precip and temp data are representing.*

We appreciate the comment but are not fully sure what additional information is requested regarding elevation, precipitation, and temperature. The details and references about precipitation, temperature, and elevation are already included in lines 95 to 100 (now from lines 98 in the new revisited manuscript version).

8.     *L100: Is there a reference of the permafrost map from South Tyrol? If not provide a short info how it was derived and clarify the relation to Boeckli et al. (2012).*

The available reference is the one indicated at line 101 (line 105 of the new manuscript version). The permafrost map of South Tyrol is based on the initial Alpine Permafrost Index (API) map developed within the PERMANENT project. This map was later refined and enhanced using results from subsequent projects such as Permaqua (https://www.permaqua.eu/de/ergebnisse.asp). The spatial resolution was improved from the original 30 m to 10 m. The mapping methodology is conceptually related to Boeckli et al. (2012) but was adapted to regional conditions using updated input data and localized calibration.

9.     *Figure 1: Include the data sources.*

Done

10.    *L108: Clarify which sensors where uses for LST extraction; also the OLI bands? TIR bands have a resolution of 100m but the OLI MS bands have 30m (as the authors are certainly aware, but it is not written as such).*

We modified the sentence as: "Using Landsat 8 Collection 2 Tier 1 data, we extracted Land Surface Temperature (LST), an Essential Climate Variable (ECV) recognized by both the Global Climate Observing System (GCOS) and the European Space Agency's Climate Change Initiative (CCI) (Galve et al., 2022; Parastatidis et al., 2017; Ermida et al., 2020). LST was derived from the thermal bands (B10 and B11) of the Thermal Infrared Sensor (TIRS), which have a native spatial resolution of 100 m. Additionally, multispectral surface reflectance bands (SR_B1 to SR_B7) and the QA_PIXEL band from the Operational Land Imager (OLI), with a spatial resolution of 30 m, were used for preprocessing tasks such as cloud masking and emissivity correction."

11.    *L122: What is the approximate spatial resolution of the ortho images?*

We added the resolution of each image: 2000 (1 m resolution), 2006 (50 cm resolution), 2008 (10 cm resolution) and 2014 (20 cm resolution).

*12.    L135: What is meant by "inner permafrost"?*

We refer to the permafrost core inside the rock glacier. We agree that it can sound redundant and, for this reason, we removed it along the manuscript.

*13.    Table 1: You may omit "table reporting the" and start with "Activity attributes…"*

Done

*14.    Table 2: Lithology and Insolation are not morphometric variables. Is the total insulation the potential or actual one? The column "Description" is not consistent; partly the impact partly the variable itself is described.*

We thank the Editor for the correction. We have modified the definition as "Geomorphological and Environmental variables" both in table 2 and in the header of section 3.2.1.
The insolation is the total one, as written in the table. We modified the "description" column to make it more consistent and avoid mixing with the impact.

| Type of variable | Parameter | Unit of measure | Description |
|---|---|---|---|
| Geomorphological and environmental variables | Lithology | categorical | Classification of surface geology by rock type |
| | Total insolation | kWh / m$^2$ | Amount of solar radiation received by a surface over a specific period |
| | Slope | ° | Angle of terrain inclination derived from elevation data |
| | Aspect | ° | Angle of the slope direction measured towards north, derived from elevation data |
| | Elevation | m.a.s.l. | Height above sea level; derived from a digital elevation model (DEM) |
| | Vector Ruggedness Measure (VRM) | / | Index quantifying terrain ruggedness based on variation in slope and aspect |
| | Convergence | / | Measure of terrain convergence and divergence, identifying ridges and valleys |
| | Profile Curvature | 1/m | Curvature of the land surface in the direction of the maximum slope; distinguishes convex and concave forms |
| Climatic | Land surface temperature (LST) | °C | Radiative skin temperature of the land surface, derived from thermal satellite data |
| | Precipitations | mm | Total amount of rain and snowfall, interpolated from ground weather station data |
| | Snow cover duration (SCD) | days | Number of days with snow cover |

*15.      L165: Which version of SAGA GIS?*

SAGA GIS 9.03. Added

*16.      L211: Mention the revisit time considering the failure of S1B and launch of S1C. See also L216-*

Thank you for the suggestion. Since our analysis covers the period from 2000 to 2022, some 6-day revisit acquisitions were still available prior to the failure of Sentinel-1B in December 2021. In the SBAS procedure we thus included the available 6-days pairs as well as longer baselines from 12days on. As we did not use Sentinel-1C data in our study, we prefer not to include information about this satellite to avoid potential confusion for the reader.

*17.      L294: I suggest "The boxplot…"*

Corrected

*18.      Figure 5: What are the yellow bars? Include this info in the legend for better readability. And "The boxplots show …"*

We corrected the caption as suggested and integrated the information on the yellow bars of the histogram, which shows the distribution (frequency) of the current variable's value across all observations (regardless of class).

*19.      L300ff: Be consistent with the abbreviations LST, SCD and VRM; they are commonly used, and partly written in full even though already introduced earlier in the manuscript.*

Thanks for pointing this out. We modified to keep only the abbreviations.

*20.      L304: A rock glacier shares (or rock glaciers share)*

Corrected in "active rock glaciers share".

*21.      L416: You may mention here or in the intro that rock glacier velocity was therefore introduced as an ECV. A suitable recent reference to consider here is: Kellerer-Pirklbauer et al. (2024).*

Thanks for the suggestion. We included the information in the intro, now in lines 77-79: "Notably, velocity has recently been recognized as a new Essential Climate Variable (ECV) (Kellerer-Pirklbauer et al., 2024), underscoring the need to incorporate velocity-based indicators when assessing the state of activity."

*22.    L422: You may also mention here Liu et al. (2013), who used this method already more than 10 years ago.*

We thank the Editor for the reference. We added it.

*23.    L426: You may refer here to the discussion about the InSAR limitations.*

We added the reference to the discussion section. Now line 439.

*24.    L461: Wood et al. (2025) found that highest velocities occur partly at the lower rock glacier near the front which corroborates your findings.*

We thank the Editor for the reference, we added it.

*25.    Figure 11: Add the meaning of the circle size in the legend of the figure.*

Done

*26.    L487: "debris cover" please clarify. You mean thickness of the ice-free debris; what is about the active layer thickness?*

With "debris material," we refer to both the surface active layer and the debris within the permafrost body. Although we cannot directly quantify the thickness of the active layer, we acknowledge that its composition and depth significantly influence the thermal properties of the rock glacier, modulating its insulation capacity and damping effect on the underlying permafrost. In addition, the abundance and grain-size distribution of debris within the permafrost body play a key role in controlling its deformation behavior, affecting both its plasticity and the potential phenomena of sliding along discrete shear zones.
We reformulate the sentence, now lines 502-505 as: "This dynamic interplay is further highlighted by the complex interactions between ice presence, debris material (which plays a key role both as surface insulation in the active layer and in controlling deformation within the rock glacier body), permafrost content, and external factors such as temperature and precipitation."

*27.    Figure 12: Show also satellite (or aerial images) of the same subset, so that the reader gets a better impression of the rock glaciers.*

We added images from Google Earth.

[Figure]

| Rock glacier | PAB | Bertone et al.2019 | New class | Confidence |
|---|---|---|---|---|
| I | Relict | A | A | 0.9 |
| II | n.d. | F | R | 0.6 |
| III | n.d. | A | R | 0.6 |
| IV | Active | F | T | 0.1 |
| V | Inactive | A | T | 0.1 |
| VI | Active | A | A | 0.4 |
| VII | Relict | A | R | 0.8 |

28.    *L521ff: You may also discuss the impact of the atmosphere.*

We have already addressed the influence of atmospheric effects in Section 3.3 and further discussed the potential limitations of the filtering CNN approach with respect to atmospheric phase screen (APS) effects in the discussion (see lines 571ff).

29.    *L537: What is the "GAMMA procedure"? Is it related to the GAMMA software?*

Exactly, as stated in HyP3 Product Guide: "The InSAR workflow used in HyP3 was developed by ASF using GAMMA software. The steps include pre-processing steps, interferogram preparation, and product creation."
https://hyp3-docs.asf.alaska.edu/guides/insar_product_guide/

*30.    L583f: I agree that the lithology is of minor importance for the velocity. However, the rock glacier velocity is predominantly driven by the shear horizon and not ice-deformation process (e.g. Arenson et al. 2002, Cicoira et al. 2021).*

Thank you for the comment. We agree that rock glacier velocity is predominantly driven by shear zone processes rather than ice deformation. We modified the sentence into "while lithology may play a minor role," but chose to retain the statement because some studies (e.g., Seppi et al., 2012) have noted correlations between lithological characteristics, such as fracturing properties, debris supply, and the ability of certain rocks to retain snowmelt, and rock glacier development.

*31.    L609: Be more specific with the variables; e.g. in the discussion section before you state the lithology is usually unimportant.*

Thank you for the suggestion. Rather than listing specific individual variables influencing rock glaciers activity, which we believe require more detailed investigation at the local scale due to their complex interplay, we have revised the sentence to the following:
"These local settings, characterized by the dynamic and complex interplay of geomorphological, environmental, and climatic variables, shape the dynamics of rock glaciers, resulting in varying rates of progression between different activity states."
This phrasing refers to the macro-classes defined in Table 2 and better reflects the integrated nature of these factors.

---

## Author Response (AR3)

**Egusphere-2024-1511**

Optimizing rock glaciers activity classification in South Tyrol (North-East Italy): integrating multisource data with statistical modelling

Chiara Crippa*, Stefan Steger, Giovanni Cuozzo, Francesca Bearzot, Volkmar Mair, and Claudia Notarnicola

**Editor comment:**

**Public justification (visible to the public if the article is accepted and published):**
Dear authors,
thank you for the careful reply and the revsions.
However, it seems that the track changes document does not show all changes. Hence, may I kinldy ask to provide a documents where all changes made are highlighted. Moreover, it is sometime unclear from the response if anything in the manuscript was changed. I would at least expect to add a sentence or or few lines. Please clarify.
Once this done, I can make a decision.

*Authors' reply:*
*We apologize to the editor for the inconvenience. The manuscript was thoroughly reviewed and revised accordingly. We are now uploading the correct version of the manuscript with all tracked changes clearly visible.*
*Please refer to our previous detailed response, where we addressed each comment and indicated the corresponding updated sentences along with line numbers in the revised manuscript.*
*All previous comments have been fully addressed.*